# Making Better Decision by Directly Planning in Continuous Control

**Jinhua Zhu**[1], **Yue Wang**[2], **Lijun Wu**[2],
**Tao Qin**[2], **Wengang Zhou**[1], **Tie-Yan Liu**[2], **Houqiang Li**[1]
[1]University of Science and Technology of China;
[2]Microsoft Research AI4Science
[1]`teslazhu@mail.ustc.edu.cn, {zhwg,lihq}@ustc.edu.cn`
[2]`{yuwang5,lijuwu,taoqin,tyliu}@microsoft.com`

## Abstract

By properly utilizing the learned environment model, model-based reinforcement learning methods can improve the sample efficiency for decision-making problems. Beyond using the learned environment model to train a policy, the success of MCTS-based methods shows that directly incorporating the learned environment model as a planner to make decisions might be more effective. However, when action space is of high dimension and continuous, directly planning according to the learned model is costly and non-trivial. Because of two challenges: (1) the infinite number of candidate actions and (2) the temporal dependency between actions in different timesteps. To address these challenges, inspired by Differential Dynamic Programming (DDP) in optimal control theory, we design a novel Policy Optimization with Model Planning (POMP) algorithm, which incorporates a carefully designed Deep Differential Dynamic Programming (D3P) planner into the model-based RL framework. In D3P planner, (1) to effectively plan in the continuous action space, we construct a locally quadratic programming problem that uses a gradient-based optimization process to replace search. (2) To take the temporal dependency of actions at different timesteps into account, we leverage the updated and latest actions of previous timesteps (i.e., step $1, \cdots, h - 1$) to update the action of the current step (i.e., step $h$), instead of updating all actions simultaneously. We theoretically prove the convergence rate for our D3P planner and analyze the effect of the feedback term. In practice, to effectively apply the neural network based D3P planner in reinforcement learning, we leverage the policy network to initialize the action sequence and keep the action update conservative in the planning process. Experiments demonstrate that POMP consistently improves sample efficiency on widely used continuous control tasks. Our code is released at `https://github.com/POMP-D3P/POMP-D3P`.

## 1 Introduction

Model-based reinforcement learning (RL) (Janner et al., 2019a; Yu et al., 2020; Schrittwieser et al., 2020; Hafner et al., 2021) has shown its promise to be a general-purpose tool for solving sequential decision-making problems. Different from model-free RL algorithms (Mnih et al., 2015; Haarnoja et al., 2018), for which the controller directly learns a complex policy from real off-policy data, model-based RL methods first learn a predictive model about the unknown dynamics and then leverage the learned model to help the policy learning. With several key innovations (Janner et al., 2019a; Clavera et al., 2019), model-based RL algorithms have shown outstanding data efficiency and performance compared to their model-free counterparts, which make it possible to be applied in real-world physical systems when data collection is arduous and time-consuming (Moerland et al., 2020).

There are mainly two directions to leverage the learned model in model-based RL, though not mutually exclusive. In the first class, the models play an auxiliary role to only affect the decision-making by helping the policy learning (Janner et al., 2019b; Clavera et al., 2019). In the second class, the model is used to sample pathwise trajectory and then score this sampled actions (Schrittwieser et al.,

2020). Our work falls into the second class to directly use the model as a planner (rather than only help the policy learning). Some recent papers (Dong et al., 2020; Hubert et al., 2021; Hansen et al., 2022b) have started walking in this direction, and they have shown some cases to support the motivation behind it. For example, in some scenarios (Dong et al., 2020), the policy might be very complex while the model is relatively simple to be learned.

These idea is easy to be implemented in the discrete action space where MCTS is powerful to do the planning by searching (Silver et al., 2016; 2017; Schrittwieser et al., 2020; Hubert et al., 2021). However, when the action space is continuous, the tree-based search method can not be applied trivially. There are two key challenges. (1) Continuous and high-dimensional actions imply that the number of candidate actions is infinite. (2)The temporal dependency between actions implies that the action update in previous timesteps can influence the later actions. Thus, trajectory optimization in continuous action space is still a challenge and lacks enough investigation.

To address the above challenges, in this paper, we propose a Policy Optimization with Model Planning (POMP) algorithm in the model-based RL framework, in which a novel Deep Differentiable Dynamic Programming (D3P) planner is designed. Since model-based RL is closely related to the optimal control theory, the high efficiency of differential dynamic programming (DDP) (Pantoja, 1988; Tassa et al., 2012) algorithm in optimal control theory inspires us to design an algorithm about dynamic programming. However, since the DDP requires a known model and a high computational cost, applying the DDP algorithm to DRL is nontrivial.

The D3P planner aims to optimize the action sequence in the trajectory. The key innovation in D3P is that we leverage first-order Taylor expansion of the optimal Bellman equation to get the action update signal efficiently, which intuitively exploits the differentiability of the learned model. We can theoretically prove the convergence rate of D3P under mild assumptions. Specifically, (1) D3P uses the first-order Taylor expansion of the optimal Bellman equation but still constructs a local quadratic objective function. Thus, by leveraging the analytic formulation of the minimizer of the quadratic function, D3P can efficiently get the local optimal action. (2) Besides, a feedback term is introduced in D3P with the help of the Bellman equation. In this way, D3P updates the action in current step by considering the action update in previous timesteps during planning. Note that D3P is a plug-and-play algorithm without introducing extra parameters.

When we integrate the D3P planner into our POMP algorithm under the model-based RL framework, the practical challenge is that the neural network-based learned model is always highly nonlinear and with limited generalization ability. Hence the planning process may be misled when the initialization is bad or the action is out-of-distribution. Therefore, we propose to leverage the learned policy to provide the initialization of the action before planning and provide a conservative term at the planning to admit the conservation principle, in order to keep the small error of the learned model along the planning process. Overall speaking, our POMP algorithm integrates the learned model, the critic, and the policy closely to make better decisions.

For evaluation, we conduct several experiments on the benchmark MuJoCo continuous control tasks. The results show our proposed method can significantly improve the sample efficiency and asymptotic performance. Besides, comprehensive ablation studies are also performed to verify the necessity and effectiveness of our proposed D3P planner.

The contributions of our work are summarized as follows: (1) We theoretically derive the D3P planner and prove its convergence rate. (2) We design a POMP algorithm, which refines the actions in the trajectory with the D3P planner in an efficient way. (3) Extensive experimental results demonstrate the superiority of our method in terms of both sample efficiency and asymptotic performance.

## 2 RELATED WORK

The full version of the related work is in Appendix A, we briefly introduce several highly related works here. In general, model-based RL for solving decision-making problems can be divided into three perspectives: model learning, policy learning, and decision-making. Moreover, optimal control theory also concerns the decision-making problem and is deeply related to model-based RL.

**Model learning:** How to learn a good model to support decision-making is crucial in model-based RL. There are two main aspects of the work: the model structure designing (Chua et al., 2018; Zhang

et al., 2021; 2020; Hafner et al., 2021; Chen et al., 2022) and the loss designing (D'Oro et al., 2020; Farahmand et al., 2017; Li et al., 2021).

**Policy learning:** Two methods are always used to learn the policy by using the learned model. One is to serve the learned model as a black-box simulator to generate the data (Janner et al., 2019b; Yu et al., 2020; Lee et al., 2020). Another way is to use the learned model to calculate the policy gradient (Heess et al., 2015b; Clavera et al., 2019; Amos et al., 2021).

**Decision-making:** When making the decision, we need to generate the actions that can achieve our goal. Many of the model-based RL methods make the decision by using the learned policy solely (Hafner et al., 2021). Similar to our paper, some works also try to make decisions by using the learned model, but the majority only focus on the discrete action space. The well-known MCTS method achieves a lot of success. For example, the well-known Alpha Zero (Silver et al., 2017), MuZero (Schrittwieser et al., 2020). There are only a few works that study the continuous action space, such as the Continuous UCT (Couëtoux et al., 2011), the sampled MuZero (Hubert et al., 2021), the TreePI (Springenberg et al., 2020), and the TD-MPC (Hansen et al., 2022a).

**Optimal control theory:** Beyond deep RL, optimal control also considers the decision-making problem but rather relies on the known and continuous transition model. In modern optimal control, Model Predictive Control (MPC) (Camacho & Alba, 2013) framework is always adopted when the environment is highly non-linear. In MPC, the action is planned during the execution by using the model, and such a procedure is called trajectory optimization. Plenty of previous works (Byravan et al., 2021; Chua et al., 2018; Pinneri et al., 2021; Nagabandi et al., 2020) use MPC framework to solve the continuous control tasks, but most of them are based on zero-order or sample-based method to do the planning. The most relevant works are DDP (Murray & Yakowitz, 1984), iLQR (Li & Todorov, 2004), and iLQG (Todorov & Li, 2005; Tassa et al., 2012). We discuss the detailed differences between our method and these methods in Appendix A.

Since our planning algorithm relies on the learned model and learned policy, we build our algorithm based on these works on **model learning** and **policy learning**. Our POMP algorithm tries to solve a more challenging task compared to the related work on **decision-making**: efficiently optimize the trajectory in continuous action space when the environment model is unknown. Different from our works, the MPC with DDP as trajectory optimizer from **optimal control theory** requires the known environment model, and also requires the hessian matrix for online optimization from scratch.

## 3 PRELIMINARIES

**Reinforcement Learning.** We consider a discrete-time Markov Decision Process (MDP) $\mathcal{M}$, defined by the tuple $(\mathcal{X}, \mathcal{A}, f, r, \gamma)$, where $\mathcal{X}$ is the state space, $\mathcal{A}$ is the action space, $f : x_{t+1} = f(x_t, a_t)$ is the transition model, $r : \mathcal{X} \times \mathcal{A} \to \mathbb{R}$ is the reward function, $\gamma$ is the discount factor. We denote the future discounted return at time $t$ as $R_t = \sum_{t'=t}^{\infty} \gamma^{t'-t} r_{t'}$, and Reinforcement Learning (RL) aims to find a policy $\pi_\theta : \mathcal{X} \times \mathcal{A} \to \mathbb{R}^+$ that can maximize the expected return $J$. where $\max_\theta J(\theta) = \max_\theta \mathbb{E}_{\pi_\theta} R_t = \max_\theta \mathbb{E}_{\pi_\theta} \left[ \sum_{t'=t}^{\infty} \gamma^{t'-t} r(x_{t'}, a_{t'}) \right]$.

**Bellman Equation.** We define the optimal value function $V^*(x) = \max \mathbb{E}[R_t | x_t = x]$. The optimal value function obeys an important identity known as the Bellman optimality equation $V^*(x) = \max_{a_t} \mathbb{E}\left[ r(x_t, a_t | x_t = x) + \gamma V^*(x_{t+1}) \right]$. The idea behind this equation is that if we know the $r(x_t, a_t)$ for any $a_t$ and next step value function $V^*(x_{t+1})$ for any $s_{t+1}$, we can recursively select the action $a_t$ which maximizes $r(x_t, a_t | x_t = x) + \gamma V^*(x_{t+1})$. Similarly, we can denote the optimal action-value function $Q^*(x, a) = \max \mathbb{E}[R_t | x_t = x, a_t = a]$, and it obeys a similar Bellman optimility equation $Q^*(x, a) = \max_{a_{t+1}} \mathbb{E}\left[ r(x_t, a_t | x_t = x, a_t = a) + \gamma Q^*(x_{t+1}, a_{t+1}) \right]$.

**Model-based RL.** Model-based RL method distinguishes itself from model-free counterparts by using the data to learn a transition model. Following Janner et al. (2019a) and Clavera et al. (2019), we use parametric neural networks to approximate the transition function, reward function, policy function and Q-value function with the following objective function to be optimized $J_f(\psi) = \mathbb{E}\left[ \log f(x_{t+1} | x_t, a_t) \right]$, $J_r(\omega) = \mathbb{E}\left[ \log r(r_t | x_t, a_t) \right]$, $J_\pi(\theta) = \mathbb{E}\left[ \sum_{t=0}^{H-1} \gamma^t r(x_t, a_t) + \right.$

$\gamma^H Q(x_H, a_H)]$ and $J_Q = \mathbb{E}\big[\|Q(x_t, a_t) - (r + \tilde{Q}(x_{t+1}, a_{t+1}))\|_2\big]$, respectively. In $J_\pi(\theta)$, we truncate the trajectory in horizon $H$ to avoid long time model rollout.

**Notations.** For one-dimensional state and action case, we denote the partial differentiation of function by using its output with subscripts, *e.g.*, $r_x \triangleq \frac{\partial r(x,a)}{\partial x}$, $r_a \triangleq \frac{\partial r(x,a)}{\partial a}$, $f_x \triangleq \frac{\partial f(x,a)}{\partial x}$, $f_a \triangleq \frac{\partial f(x,a)}{\partial a}$, $Q_x \triangleq \frac{\partial Q(x,a)}{\partial x}$ and $Q_a \triangleq \frac{\partial Q(x,a)}{\partial a}$. See Appendix E for the multi-dimension case.

## 4 PLANNING IN CONTINUOUS ACTION SPACE

In this section, we present our POMP algorithm and the D3P planner in detail. First, we derive the D3P planner which relies on the Bellman equation. Then, we theoretically prove its convergence property. Finally, we show how to effectively apply D3P planner in our POMP algorithm in RL.

### 4.1 DEEP DIFFERENTIAL DYNAMIC PROGRAMMING

In this subsection, we will theoretically derive the D3P planner and prove its convergence property. There are mainly two challenges in continuous action space planning: (1) the infinite number of candidate actions, and (2) the temporal dependency between actions in different timesteps.

Here, we briefly introduce the main idea of our D3P planner to solve the above challenges. We first define an objective function and formulate it as an optimization problem based on the Bellman equation. Then, we convert it to a local optimization problem and approximate the objective function via Taylor expansion. To avoid the computation of the hessian matrix, we use the first-order Taylor expansion to construct a quadratic function. Since the analytical solution of a quadratic function is easy to get, we can efficiently get the local optimal action sequence and thus overcome the challenge (1) to some extent. To get over challenge (2), we introduce a feedback term into the objective function to depict the state change induced by the action update in prior timesteps. By considering the feedback term that explicitly involves the information of prior action updates, we can correct the action update in time. The remaining question is whether the D3P planner can indeed optimize the original objective after we make several approximations when deriving the algorithm. Through theoretical analysis, we show that the convergence rate of the proposed algorithm can be guaranteed.

We now introduce how we derive the D3P planner. For clarification, we use the finite horizon MDP as a proof of concept setting. The state and action are one-dimensional variables. The infinite horizon MDP with multi-dimensional state and action can be derived similarly and we put it in Appendix E. Recall the goal of RL methods, our planning algorithm aims to find the action sequences $\{a_1, \cdots a_H\}$ that can maximize the value function $V(x_1, 1) \triangleq \max_{a_1, \cdots a_H} \sum_{h=1}^{H} r(x_h, a_h)$, where $x_{h+1} = f(x_h, a_h)$.

Due to challenge (1), such an optimal action sequence is in general hard to find. Hence our D3P planner treats this optimal action sequence searching problem as an optimization problem that leverages the optimal Bellman equation to formulate the following objective function,

$$V(x_h, h) = \max_{a_h}[r(x_h, a_h) + V(f(x_h, a_h), h + 1)]. \tag{1}$$

Since the reward function and the transition function is unknown, we will use neural network to approximate them. However, the optimization problem is highly non-convex. Thus, we consider an auxiliary goal that is to find the local optimal $a + \delta a$ in the neighbourhood of current action $a$ to improve the action from $a$ to $a + \delta a$. Denote $Q(x_h, a_h) = r(x_h, a_h) + V(f(x_h, a_h), h + 1)$, our goal can be re-expressed as $\delta a_h = \arg\max_{\delta a}[Q(x_h, a_h + \delta a)]$.

To accelerate the optimization process, D3P planner constructs a quadratic objective function to get the local optimal action analytically. Specifically, we propose to use the first-order Taylor expansion to avoid computing the hessian matrix. However, the first-order Taylor expansion can not lead to a quadratic objective function directly, hence we first seek a surrogate objective function $D(x, a) \triangleq (Q(x, a) - V_{max})^2$, where $V_{max}$ is a constant and set to larger than the upper bond of $Q(x, a)$. It is easy to check that $\arg\min_{\delta a} D(x, a + \delta a) \triangleq \arg\max_{\delta a} Q(x, a + \delta a)$.

For challenge (2), intuitively, after updating the action $a_t$ in prior timestep, state $x_{t+1}$ will change and we should update the action $a_{t+1}$ accordingly. Such a manner is often called "feedback". To

---

**Algorithm 1** Deep Differential Dynamic Programming (D3P)

---

**Require:** Initial action sequences $\{a_t\}_{t=1\cdots H}$, initial state $x_1$, iteration number $N_d$, valid horizon $H$, maximum expected improvement $V_{max} - Q(x, a)$.

1: **for** $i = 1, \cdots, H$ **do**             # Initialize the trajectory.
2:   Calculate $r_i = r_\omega(x_i, a_i), x_{i+1} = f_\psi(x_i, a_i)$.
3: **end for**
4: **for** $i = 1, \cdots, N_d$ **do**            # Optimize the trajectory.
5:   Calculate $Q_x(x_H, a_H), Q_a(x_H, a_H)$.       # Backward process.
6:   **for** $j = H - 1, \cdots, 1$ **do**
7:    Calculate $r_a, r_x, f_a, f_x$.
8:    Calculate $Q_a, Q_x, k, K, V_x$ using Equation 3, 4, 5 and 9.
9:   **end for**
10:   $\delta x_1 = 0$.               # Forward process.
11:   **for** $j = 1, \cdots, H$ **do**
12:    Calculate $\delta a_j$ using Equation 3, and $a_j \leftarrow a_j + \delta a_j$.
13:    Calculate $x_{j+1} \leftarrow f_\psi(x_j, a_j)$, and $\delta x_{j+1} = x_{j+1} - x_j$.
14:   **end for**
15: **end for**
16: **return** The last best action $a_1$.

---

achieve the feedback control, we now consider $Q(x + \delta x, a + \delta a)$, in which $\delta x$ represents the state change due to the prior action update. Applying first-order Taylor expansion for the Q function in D function we can get a quadratic function of $\delta a$(recall the notations in Preliminary)

$$\tilde{D}(x + \delta x, a + \delta a) = (Q(x, a) + Q_a(x, a)\delta a + Q_x(x, a)\delta x - V_{max})^2. \tag{2}$$

we now get the optimal action update $\delta a^*$ as a function of the feedback $\delta x$, denote $k_h = \frac{Q(x_h, a_h) - V_{max}}{Q_a(x_h, a_h)}$ and $K_h = \frac{Q_x(x_h, a_h)}{Q_a(x_h, a_h)}$,

$$\delta a_h^* = -k_h - K_h \delta x_h = -\frac{Q(x_h, a_h) - V_{max}}{Q_a(x_h, a_h)} - \frac{Q_x(x_h, a_h)}{Q_a(x_h, a_h)}\delta x_h. \tag{3}$$

The remaining part is how to calculate the $Q_x(x, a), Q_a(x, a)$ in the update rule,

$$Q_a(x_h, a_h) = r_a(x_h, a_h) + V_x(f(x_h, a_h), h + 1) \cdot f_a(x_h, a_h), \tag{4}$$
$$Q_x(x_h, a_h) = r_x(x_h, a_h) + V_x(f(x_h, a_h), h + 1) \cdot f_x(x_h, a_h). \tag{5}$$

By leveraging the differentiable model including the reward and transition function, only the gradient of value function $V_x(f(x_h, a_h), h + 1)$ is hard to calculate. We use the Bellman equation and Taylor expansion once again to calculate $V_x(f(x_h, a_h), h + 1)$. Putting $\delta a_h^*$ into Bellman equation (1) and using Taylor expansion ,

$$V(x_h + \delta x_h, h) = Q(x_h + \delta x_h, a_h + \delta a_h^*) \tag{6}$$
$$\approx Q(x_h, a_h) + Q_x(x_h, a_h)\delta x_h + Q_a(x_h, a_h)\delta a_h^* \tag{7}$$
$$= \underbrace{(Q(x_h, a_h) - Q_a(x_h, a_h)k_h)}_{\text{zero-order term}} + \underbrace{(Q_x(x_h, a_h) - Q_a(x_h, a_h)K_h)\delta x_h}_{\text{first-order term}}. \tag{8}$$

We can now use the coefficient of the first-order term in Taylor expansion of $V(x_h + \delta x_h, h)$ to calculate the $V_x$

$$V_x = Q_x(x_h, a_h) - Q_a(x_h, a_h)K_h. \tag{9}$$

The whole D3P planner is shown in Algorithm 1. Noting that the current presentation of our method is applied in the deterministic environment, but our D3P planner can be easily extended to the stochastic environment with reparameterization tricks (such as normal distribution noise in Kingma & Welling (2013)). Since we adopt some approximation in the derivation of the algorithm, we need some convergence guarantee.

**Theorem 1.** *Let $\{x_h, a_h\}_{h=1,\cdots,H}$, denote the current state and action in a sequence of length T. Let $\{a_h' = a_h + \delta a_h\}_{h=1,\cdots,H}$ denote the new actions updated once by D3P planner. Under mild*

*assumptions, we can prove that for $h \in \{1, \cdots, H\}$, there exist constant $C$ and $B$ such that*

$$\|a_h' - a_h^*\| \leq C \sum_{k=1}^{H} \|a_k - a_k^*\|^2 + B \sum_{k=1}^{H} \|a_k - a_k^*\|, \tag{10}$$

*where $C$ proportional to the Lipschitz (denoted $L_1$) and smoothness (denoted $L_2$) constant of the transition function and reward function $C = \mathcal{O}(L_1, L_2)$, $B$ proportional to the scale of the second order derivation of the transition and reward function $B = \mathcal{O}(f_{aa}, f_{ax}, f_{xx}, r_{aa}, r_{ax}, r_{xx})$.*

The above theorem shows that if we can choose a good initialization point for the planning process, we can guarantee the asymptotic convergence of the planning process. For the finite sample case, the convergence rate is at least linear convergence. If the second derivative of the transition function is near zero ($B$ is sufficient small), the convergence rate is near quadratic convergence. The intuition is shown in Lemma 2. In this situation, the 2nd order derivative of D can be approximated by the multiplication of the 1st order derivative of $Q$ and thus of $f$ and $r$. For example $D_{aa} \approx Q_a Q_a$ .

We further analyze the influence of the feedback term in terms of the convergence rate.

**Corollary 1.** *If we do not consider the feedback term ($\delta x = 0$), the convergence rate is $\|a_h' - a_h^*\| \leq C \sum_{k=1}^{H} \|a_k - a_k^*\|^2 + B \sum_{k=1}^{H} \|a_k - a_k^*\| + \frac{Q_x(x_h, a_h)}{Q_a(x_h, a_h)} \sum_{i=h-1}^{1} \Pi_{j=i+1}^{h-1} f_x(x_j, a_j) \left[ f_a(x_i, a_i)\delta a_i + C\delta a_i^2 \right].$*

The corollary shows that if we do not consider the temporal dependency between actions in different timestep, or in other words $\delta x = 0$, the convergence rate will be slower than Equation (12) with an extra error term. The intuition is, since we are optimizing the action sequence along a trajectory, the action update will change the trajectory. Given our objective is a function of state and action, the different states will lead to the different optimal actions. Therefore, if we do not consider the state change due to the action update in the previous timesteps, the action update direction will not be toward the true gradient direction. Besides, the influence is proportional to the magnitude of the state change which is determined by the system property ($f_x, f_a$) and previous action update $\delta a_i$.

## 4.2 POLICY OPTIMIZATION WITH MODEL PLANNING: A PRACTICAL IMPLEMENTATION

In this subsection, we show how we apply our D3P planner to the deep RL framework. Since the D3P planner is a plug-and-play algorithm, compared to the traditional model-based RL algorithm like MAAC (Clavera et al., 2019), only the decision-making parts are different. The POMP algorithm is summarized in Appendix B. Note that D3P planner module does not introduce any additional neural networks. All network structure, including model, critic, and policy are the same as MAAC (Clavera et al., 2019) and MBPO (Janner et al., 2019b).

One key problem that needs to be resolved before applying the D3P planner is how to avoid misleading planning due to the limited generalization ability of the learned model. Such a problem can not be ignored as long as the ground-truth model is unknown, which can only be learned by data with function approximation. We consider two components in the algorithm to alleviate the effect of the model error: the initialization strategy and the conservative planner objective.

For the initialization strategy, we propose to use the policy network and learned model to initialize the state-action trajectory. That is, the initial action used by D3P planner is the output of the learned policy. The motivations are as follows. (1) Since the policy is trained to maximize the return-to-go as general model-based RL, the proposed action would be reasonable and competitive, which is better than random initialization. (2) Since the data used to train policy is sampled from the replay buffer, the action outputted by the policy network should lead to a small model prediction error.

For the conservative planner objective, constraining the actions outputted by D3P planner near the training data can keep the model prediction error small and provide an additional regularization for the planner. Specifically, since the policy output is a multivariate Gaussian, we can easily calculate the log-likelihood $\log P(x_i, a_i)$ for a given state action pair. The log-likelihood is used as an auxiliary reward, and we add it to the output of the reward function when doing planning in the evaluation phase. Specifically, we add an additional reward at the first step, and the optimization objective of D3P becomes $J_c(\{a_i, \cdots, a_{i+H-1}\}) = \sum_{h=i}^{i+H-2} r(x_h, a_h) + Q(x_{i+H-1}, a_{i+H-1}) + \alpha \log P(x_i, a_i)$, where $\alpha$ is a hyper-parameter. Please note that we only use this conservative term during evaluation, as we want to encourage exploration when training.

## 5  EXPERIMENTS

In this section, we aim to answer the following questions: (1) Compared to state-of-the-art methods, how does our method perform on benchmark continuous control tasks? (2) Is planning necessary to make a better decision in continuous control? (3) Is our D3P planner advantageous in continuous control? (4) How the learned model quality affects decision-making? (5) Does our D3P efficiently optimize the trajectory quality? (6) Is the policy network necessary in our framework? To answer the above questions, we evaluate our method on continuous control benchmark tasks in the MuJoCo simulator (Todorov et al., 2012). Our method is built on top of MAAC(Clavera et al., 2019), which means the procedure of model learning, policy optimization, and the corresponding hyper-parameters are the same as MAAC. More details are left in Appendix C.3. Due to space limitation, we leave the detailed description of the baseline methods in Appendix C.4.

### 5.1  COMPARISONS WITH EXISTING METHODS

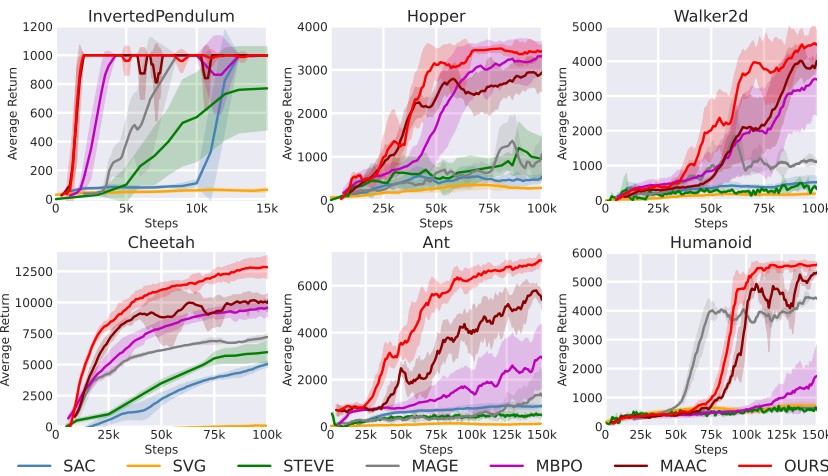

Figure 1:   Learning curves of our method and other baseline methods on six continuous control tasks. The solid lines represent the mean of 10 (for our method)/5 (for other baseline methods) trails with different seeds, and the shaded regions correspond to STD among trials. Our method achieves the best results among these strong model-free and model-based reinforcement learning methods.

To answer the first question, we compare our method with six SOTA baseline methods, and the results are shown in Fig. 1. Specifically, no matter on asymptotic performance or on the sample efficiency, our method shows a significant performance improvement against MAAC, of which our method is built on top, on all six tasks. Moreover, on two control tasks with high-dimensional action space, Ant and Humanoid, the improvement of our method are more obvious. In general, our method achieves better performance than all other model-based and model-free baseline methods, which demonstrates the effectiveness and generality of our method. Note that in humanoid task, MAGE achieves better sample efficiency than ours in early training phase, but our method achieves a better final result than MAGE and MAGE is worse than our method on all other tasks.

### 5.2  ABLATION STUDIES

In this section, we conduct several ablation experiments to answer questions (2) (6) posted before and show the necessity and effectiveness of the proposed components in our method.

**Is planning necessary to make a better decision in continuous control?** We design experiments to verify the effectiveness of two possible ways to make a better decision: (1) Using the model to do planning and (2) Increasing the $N_p$ in Algorithm 2, which is the number of update times of the policy net after we collect 1 data from the real environment, and then relying on the policy to make the decision. Here we increase $N_p$ from 10 (in MAAC original implementation) to $\{20, 50, 100\}$ to see whether increasing the update times of the policy could help policy optimization, and the results

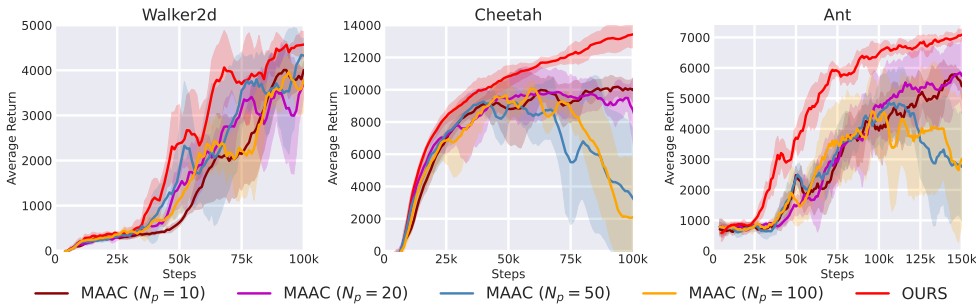

Figure 2: Ablation about the update times $N_p$ of policy in each iteration. We can see that increasing $N_p$ cannot help policy optimization.

are presented in Figure 2. As shown in the figure, $N_p = 10$ in the original MAAC is a rather good choice, and increasing $N_p$ even would harm the policy optimization. However, our method, which uses the learned model as a planner could consistently improve the policy.

**Is our D3P planner advantageous in continuous control?** D3P planner considers the temporal dependency and constructs a local quadratic objective function to optimize the initial trajectory proposed by the policy network. To validate the advantage of our method, we replace the D3P planner in our method with an SGD-like planner, which directly optimizes the action sequence with gradient ascend; a random-shooting planner (Press et al., 2007), which randomly samples some actions in the entire action space and then scores these actions according to the reward and critic function; a cross-entropy method (CEM) planner (Rubinstein & Kroese, 2004; Hansen et al., 2022a), which adaptively and iteratively adjusts the sampling distribution in a sophisticated manner. Noting that we only change the planner in all these variants, and keep the model and policy learning unchanged for a fair comparison. The results are shown in Figure 3, and we can see that SGD-like planner (denoted by POMP with SGD planner) performs similarly to policy network (denoted by MAAC) and the improvement over policy (MAAC) is limited. Our method (denoted by POMP with D3P planner) is more effective than SGD-like planner. Moreover, the gaps between our method and the CEM planner (denoted by CEM), the random-shooting planner (denoted by Random-shooting) clearly show the efficiency of the first-order method (compared to the zero-order method).

**How the learned model quality affect decision-making?** As our method optimizes the trajectory via planning in a learned environment model, a key part is to see how the learned model quality affects the planning results. To answer this question, we pick 4 types of the learned model with different amount of training data ( the more training data, the better the quality of the learned model). Then we cluster the policy network according to their performance into 6 groups. Finally, we combine the different quality models with each policy group to see the average performance improvement after we applying the D3P planner on the learned model and policy. First, for each

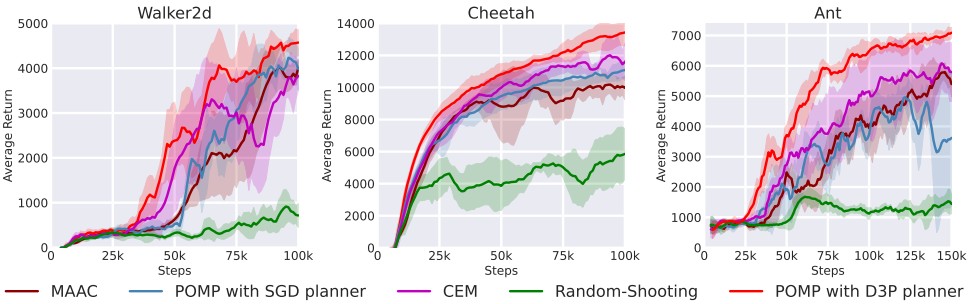

Figure 3: Ablation studies about D3P planner. We replace the D3P planner in our method with a SGD-like planner, a CEM planner, and a random-shooting planner, the results show the advantage of our D3P planner.

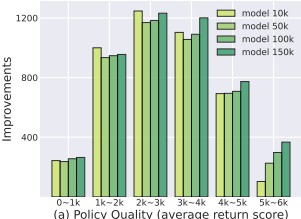 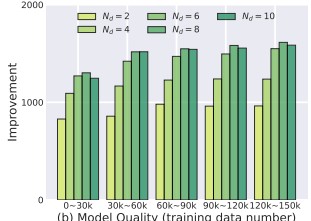 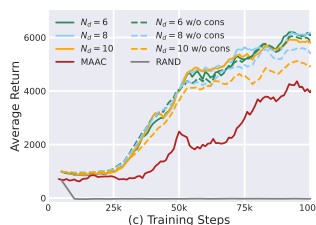

Figure 4: (a) The improvement of applying learned model with different training steps on policy with different quality. "Improvement" means the evaluation return using our D3P planner to subtract the return that without our D3P planner. "Policy quality" means the average episode return of the policy when applying the policy in the environment, and "$ik{\sim}(i+1)k$" denotes the policy cluster whose average return lies in $ik{\sim}(i+1)k$. "Model $ik$" denotes the learned model which is trained using $ik$ data. (b) The improvement of different iteration number $N_d$ in D3P (Line 4 in Algorithm 1). "Model quality" means the number of training data used to train the model, and "$ik{\sim}jk$" denotes the learned model with $ik{\sim}jk$ training data. (c) Ablation about the policy usage in our method. "RAND" denotes POMP with a randomly initialized trajectory rather than a policy generated trajectory in D3P. "$N_d = i$" denotes POMP with iteration number $i$ and "$N_d = i$ w/o cons" denotes POMP with iteration number $i$ and without the conservative term when evaluation.

model and each policy, we evaluate the average return using 10 trajectories. Then, we cluster the learned model and policy according to their training data and the average return and then calculate the average performance improvement in each cluster. From the result shown in Figure 4(a): (1) the improvement of the model trained on only $10k$ train data is similar to those of models trained by more data (except $5k{\sim}6k$ is slightly worse), which means it is enough to use an early stage model in our D3P planner; (2) our D3P planner could consistently improve the performance of the decision made by policy network directly, especially in early and middle stage.

**Does our D3P efficiently optimize the trajectory quality?** Similarly, we cluster the learned model according to their used training data, and combine it with a fixed policy (with an average return about $4k$) and see the impact of different iteration numbers $N_d$ used in our D3P planner. From the results shown in Figure 4(b): (1) the performance improvements increase as we use more iteration numbers, which shows the effectiveness of our method; (2) the improvements are almost the same after $N_d >= 6$, , and we do not need more iterations, which demonstrate the efficiency of our method; (3) the results also show that the early stage model is enough for our D3P planner.

**Is the policy network necessary in our framework?** There are two usages for the policy network in our D3P planner: (1) initialize the trajectory to be optimized, (2) add a conservative term as an auxiliary reward during evaluation. We conduct an ablation experiment to verify the necessity of the policy network in our method, and the results are shown in Figure 4(c). First, when we use a trajectory randomly generated rather than proposed by a policy network, the D3P failed to find any meaningful action (denoted by "RAND"), which proves the importance of trajectory initialization. Second, as we increase the iteration number in D3P planner, the performance with our conservative term is consistently better than those without it, especially at the later stage when the policy network is near optimal. This means the generality of the learned model is limited when we use a large iteration number $N_d$, and we need to constrain the optimization space of the method.

## 6 CONCLUSIONS AND FUTURE WORK

In this work, we first derived the D3P planner which is effective and efficient for continuous control and proved its convergence rate. Then, we proposed the POMP algorithm, which leverages our D3P planner in a practical model-based RL framework. Extensive experiments and ablation studies on benchmark continuous control tasks demonstrate the effectiveness of our method and show the benefit of utilizing the model planning in continuous control. For future work, given the model uncertainty can effectively trade-off the exploration and exploitation, how to properly estimate and incorporate the uncertainty of the learned model into planning is a meaningful topic.

ACKNOWLEDGMENTS

This work was supported in part by NSFC under Contract 61836011, and in part by the Fundamental Research Funds for the Central Universities under contract WK3490000007.

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

# A    RELATED WORK

Model-based RL methods for solving decision-making problems focus on three key perspectives: how to learn the model? how to use the learned model to learn the policy? And how to make the decision using the learned model and policy? Besides, decision-making that relies on the model is also investigated in the optimal control theory field which is deeply related to model-based RL.

**Model learning:** How to learn a good model to support decision-making is a crucial problem in model-based RL. There are two main aspects of the work: the model structure designing and the loss designing. For model structure designing, ensemble-based model (Chua et al., 2018), dropout mechanisms (Zhang et al., 2021), auto-regressive structure (Zhang et al., 2020), stochastic hidden model (Hafner et al., 2021), and transformer based model (Chen et al., 2022) are always considered to improve the model robustness and prediction accuracy. For loss designing,  decision awareness (D'Oro et al., 2020; Farahmand et al., 2017) and gradient awareness (Li et al., 2021) are always considered to reduce the gap between model learning and model utilization.

**Policy learning:** Two methods are always used to learn the policy by using the learned model. One is to serve the learned model as a black-box simulator to generate the data. Janner et al. (2019b) is a representing work of this line. Yu et al. (2020), Lee et al. (2020) also follow such a manner by extending it to offline-RL setting.  Another way is to use the learned model to calculate the policy gradient. Heess et al. (2015b) presents an algorithm to calculate the policy gradient by back-propagating through the model. Clavera et al. (2019) and Amos et al. (2021) share similar methods but use promising actor and critic learning strategy to achieve better performance.

**Decision-making:** When making the decision, we need to generate the actions that can achieve our goal. Most of the model-based RL methods make the decision by using the learned policy solely (Janner et al., 2019b; Yu et al., 2020; Clavera et al., 2019; Hafner et al., 2021). Similar to our paper, some works also try to make decisions by using the learned model, but the majority only focus on the discrete action space. For example, the well-known Alpha Zero system (Silver et al., 2017) uses MCTS to derive the action by using the known model. In MuZero and  (Schrittwieser et al., 2020), the authors propose to use a learned model combined with an MCTS planner to achieve significant performances in a broad range of tasks within discrete action space.  There are only a few works that study the continuous action space. Couëtoux et al. (2011) extends the MCTS framework to continuous action space but also needs to know the real model and handle the model. In Hubert et al. (2021), the author proposed a sampled MuZero algorithm to handle the complex action space by planning over sampled actions. In Hansen et al. (2022a), the authors propose to learn a value function that can be used as long term return in the Cross-Entropy (CE) method for planning.

**Optimal control:** Beyond deep RL, optimal control also considers the decision-making problem but rather relies on the known and continuous transition model. In modern optimal control theory, Model Predictive Control (MPC) (Camacho & Alba, 2013) framework is always adopted when the environment is highly non-linear. In MPC, the action is planned during the execution by using the model, and such a procedure is called trajectory optimization.  There are plenty of previous works that use the MPC framework to solve continuous control tasks. For example, Byravan et al. (2021) proposes to use sampling-based MPC for high-dimensional continuous control tasks with learned models and a learned policy as a proposal distribution. Pinneri et al. (2021) proposes an improved version of the Cross-Entropy Method for efficient planning. Nagabandi et al. (2020) proposes a PDDM method that uses a gradient-free planner algorithm combined with online MPC method to learn flexible contact-rich dexterous manipulation skills.

**Differential Dynamical Programming:** The most relevant works are DDP (Murray & Yakowitz, 1984), iLQR (Li & Todorov, 2004), and iLQG (Tassa et al., 2012). Differentiable Dynamic Programming (DDP) (Tassa et al., 2012) employs the Bellman equation structure (Murray & Yakowitz, 1984; Pantoja, 1988; Aoyama et al., 2021) and has fast convergence property.  It becomes more and more popular in the control field. iLQR (Li & Todorov, 2004), and iLQG (Tassa et al., 2012; Todorov & Li, 2005) are two variants of the DDP. In iLQR and iLQG, the second-order derivative of the environment model is ignored (set as zero). Therefore, iLQR and iLQG are more computationally efficient compared to the original DDP method. Since both iLQG and our D3P planner are motivated by DDP, they look similar naturally.  But our method has several key differences compared with theirs, and these differences are well-designed to incorporate the neural network model. (1) DDP, iLQR, and iLQG are both pure planning algorithms that require a known environment

model. (2) Computing the second-order derivative of the neural network based model is computationally costly (Hessian matrix). In our method, we only rely on the first-order derivative of the model. (3) The previous methods use the second-order Talyor expansion of the Q-value function to handle the local optimization problem. But it is hard to guarantee that the hessian matrix is a negative definite matrix, which is a necessary condition for convergence. Here, we construct an auxiliary target function $D$ and use a first-order Talyor expansion for the $Q$ function inside of the $D$ function to guarantee the non-positive definite matrix.

## B  POMP ALGORITHM

In this subsection, we present the details of POMP algorithm. Overall speaking, POMP algorithm learn three components: model, critic, and actor with the neural network function approximator, and leverage the D3P planner module to integrate all three components to make a better decision.

The POMP algorithm runs as follows. We first learn the model, the policy, and the critic using pre-given or random-policy-generated data. Then, we leverage the D3P planner to generate actions based on the model, the critic, and the policy network to interact with the environment and add these data to the true replay buffer. Next, we will use the data from the true replay buffer to train the model. We also generate fake data by using the learned model and add these data to the fake replay buffer. After that, we will sample the data from both real buffer and fake buffer to train the critic and policy. We will repeat the training process until certain convergence conditions are satisfied. When doing planning and rollout with the learned model to generate fake data, we follow the method used by Janner et al. (2019a); Clavera et al. (2019) to truncate the trajectory and use Q-function to approximate the return after the truncation. When updating the policy, we calculate the policy gradient by back-propagating through the model which is inspired by Clavera et al. (2019). When updating the critic, we follow the SAC (Haarnoja et al., 2018) to construct two Q-functions with two target Q-functions and apply the soft Q-update.

---

**Algorithm 2** POMP

---

**Require:** Policy update times $N_p$, total interaction number $N$, model train frequency $k$.
 1: Initialize the learnable model $f_\psi$, the reward function $r_\omega$, the policy network $\pi_\theta$, the critic $Q_\phi$, true replay buffer $\mathcal{D}_{env} \leftarrow \emptyset$, fake replay buffer $\mathcal{D}_{model} \leftarrow \emptyset$.
 2: **for** $i = 1, \cdots, N$ **do**
 3:     Initialize the action sequence using policy net $\pi_\theta$, and learned model $f_\psi$.
 4:     Interact with real environment $\mathcal{E}_{real}$ using D3P planner (Algorithm 1), and add the transition to $\mathcal{D}_{env}$.
 5:     **if** $i \mod k == 0$ **then**
 6:         **repeat**
 7:             Update $\psi \leftarrow \psi - \alpha_f \nabla_\psi J_f, \omega \leftarrow \omega - \alpha_r \nabla_\omega J_r$ using data from $\mathcal{D}_{env}$.
 8:         **until** The learnable model and reward function converge.
 9:     **end if**
10:     Sample transitions with $f_\psi$, and add them to $\mathcal{D}_{model}$.
11:     $\mathcal{D} \leftarrow \mathcal{D}_{env} \cup \mathcal{D}_{model}$
12:     **for** $j = 1, \cdots, N_p$ **do**
13:         Update $\theta \leftarrow \theta - \alpha_\pi \nabla_\theta J_\pi$ using data from $\mathcal{D}$.
14:         Update $\phi \leftarrow \phi - \alpha_Q \nabla_\phi J_Q$ using data from $\mathcal{D}$.
15:     **end for**
16: **end for**
17: **return** Optimal parameters $\psi^\star, \omega^\star, \theta^\star$ and $\phi^\star$.

---

## C  EXPERIMENT SETUP

### C.1  IMPLEMENTATION DETAILS

**How to set $V_{max} - Q(x, a)$?** In our D3P method, we introduce a constant $V_{max}$ and set it larger than the upper bound of $Q(x, a)$. However, we can not know the true value of the upper bound of $Q(x, a)$, and setting a too large or small $V_{max}$ is not perfect for planning. In our implementation, we fist define

| | InvertedPendulum | Hopper | Walker2d | Cheetah | Ant | Humanoid |
|---|---|---|---|---|---|---|
| Training Steps | 15000 | 100000 | 100000 | 100000 | 150000 | 150000 |
| Batch Size | 256 | 256 | 256 | 256 | 256 | 256 |
| Learning Rate | $3e-4$ | $3e-4$ | $3e-4$ | $3e-4$ | $3e-4$ | $3e-4$ |
| Horizon | 4 | 3 | 4 | 4 | 4 | 4 |
| $N_p$ | 10 | 10 | 10 | 10 | 10 | 10 |
| $N_d$ | 10 | 10 | 10 | 10 | 10 | 10 |
| Model train freq $k$ | 250 | 250 | 250 | 250 | 250 | 250 |
| Ensemble Size | 7 | 7 | 7 | 7 | 7 | 7 |
| Maximum $V_{max} - Q(x,a)$ | 50 | 20 | 10 | 20 | 20 | 50 |

Table 1: The detailed hyper-parameters in our experiments.

a maximum expected improvement $\Delta$ and then grid search $V_{max} - Q(x,a) := \{\exp\left(\frac{\log \Delta}{K} \times i\right)|i = 1, \cdots, K\}$ to get the best $V_{max}$ according to our planning objective function. Please note that the grid search are implemented in parallel.

## C.2 DESCRIPTIONS OF OUR EXPERIMENT ENVIRONMENTS

Following prior model-based RL work, we conduct our experiments on 6 classical continuous control tasks from MuJoco (Todorov et al., 2012), and the descriptions of these environments are summarized as follows[1]:

1. **Inverted Pendulum**: This environment involves a cart that can be moved linearly, with a pole fixed on it at one end and having another end free. The cart can be pushed left or right, and the goal is to balance the pole on the top of the cart by applying forces on the cart. The action space dimension and state space dimension are 1 and 4, respectively.

2. **Hopper**: The hopper is a two-dimensional one-legged figure that consists of four main body parts - the torso at the top, the thigh in the middle, the leg in the bottom, and a single foot on which the entire body rests. The goal is to make hops that move in the forward (right) direction by applying torques on the three hinges connecting the four body parts. The action space dimension and state space dimension are 3 and1 11, respectively.

3. **Walker2D**: The walker is a two-dimensional two-legged figure that consists of four main body parts - a single torso at the top (with the two legs splitting after the torso), two thighs in the middle below the torso, two legs in the bottom below the thighs, and two feet attached to the legs on which the entire body rests. The goal is to make coordinate both sets of feet, legs, and thighs to move in the forward (right) direction by applying torques on the six hinges connecting the six body parts. The action space dimension and state space dimension are 6 and 17, respectively.

4. **Half Cheetah**: The HalfCheetah is a 2-dimensional robot consisting of 9 links and 8 joints connecting them (including two paws). The goal is to apply a torque on the joints to make the cheetah run forward (right) as fast as possible, with a positive reward allocated based on the distance moved forward and a negative reward allocated for moving backward. The torso and head of the cheetah are fixed, and the torque can only be applied on the other 6 joints over the front and back thighs (connecting to the torso), shins (connecting to the thighs), and feet (connecting to the shins). The action space dimension and state space dimension are 6 and 17, respectively.

5. **Ant**: The ant is a 3D robot consisting of one torso (free rotational body) with four legs attached to it with each leg having two links. The goal is to coordinate the four legs to move in the forward (right) direction by applying torques on the eight hinges connecting the two links of each leg and the torso (nine parts and eight hinges). The action space dimension and state space dimension are 8 and 27, respectively.

6. **Humanoid**: The 3D bipedal robot is designed to simulate a human. It has a torso (abdomen) with a pair of legs and arms. The legs each consist of two links, and so do the arms

---

[1]Please refer to `https://www.gymlibrary.dev/environments/mujoco/` for more details.

(representing the knees and elbows respectively). The goal of the environment is to walk forward as fast as possible without falling over. The action space dimension and state space dimension are 17 and 376, respectively.

### C.3 EXPERIMENTAL DETAILS

In our method, for a fair comparison, except the D3P planning, we keep the model learning, policy learning, and Q-function learning to be the same as prior work (Janner et al., 2019b; Clavera et al., 2019). Specifically, the learnable prediction model is parameterized by an ensemble of 7 individual 5-layer MLPs, and is trained by Adam optimizer (Kingma & Ba, 2014) with all history transition data from replay buffer after certain hundreds of timesteps (the timesteps vary depending on the specific task); after each interaction with the environment, the policy is optimized using the pathwise derivative of the imagined trajectory produced by the learned model and the learned policy; the Q-function is learned by minimizing the TD-error for each history data saved in replay buffer and imagined data from learned model and policy function. The detailed hyper-parameters are summarized in Table 1, and refer to Janner et al. (2019b); Clavera et al. (2019) for more details.

Noting that our planner is built upon the framework of MBPO and MAAC. Therefore, the sample efficiency of our method is comparable with MBPO and MAAC which also used the same state augmentation strategy. So, the improvement of the sample efficiency is not relevant to the state augmentation strategy.

### C.4 THE DESCRIPTION OF BASELINE METHODS

To show the effectiveness of our algorithm, we compare our method on six classical continuous control tasks against the following state-of-the-art model-free and model-based RL algorithms: (i) Soft Actor-Critic (SAC) (Haarnoja et al., 2018), a popular off-policy actor-critic RL algorithm based on maximum entropy RL framework; (ii) SVG(1) (Heess et al., 2015a), which first uses dynamics derivatives in model-based RL; (iii) STochastic Ensemble Value Expansion (STEVE) method (Buckman et al., 2018), which utilizes the learned models only when the uncertainty of the learned model is not too high; (iv) Model-based Action-Gradient-Estimator policy optimization (MAGE) method (D'Oro & Jaśkowski, 2020), which computes gradient targets in temporal difference learning by backpropagating through the learned dynamics; (v) Model-Based Policy Optimization (MBPO) method (Janner et al., 2019b), which shows that using short model-generated rollouts branched from real data could benefit model-based algorithms; (vi) Model-Augmented Actor-Critic (MAAC) (Clavera et al., 2019) method, which exploits the learned model by computing the analytic gradient of the returns with respect to the policy.

## D MORE EXPERIMENTAL RESULTS

### D.1 STUDIES ON THE ROBUSTNESS OF OUR METHOD.

We test the sensitivity of our method when we change the hyperparameter used in the training phase. The ablation studies about iteration number $N_d$ used in our training phase and the maximum expected improvement $V_{max} - Q(x, a)$ (which we denote by $\Delta$) are shown in Figure 5 and Figure 6, respectively. We can see that our method consistently outperforms MAAC, and the hyper-parameter choice is not much sensitive to our method.

### D.2 COMPARISON WITH CONTINUOUS MUZERO

MuZero (Schrittwieser et al., 2020) is a successful model-based RL method for discrete action tasks, which carefully trades off the exploitation and exploration. To compare these tree-based methods with our gradient-based method, we conduct a comparison with MuZero combined with Continuous UCT (Progressive Widening (Couëtoux et al., 2011) in our experiments[2]. We gird search several important hyper-parameters for the continuous MuZero variant, and the detailed hyper-parameters are summarized in Table 2. The results are shown in Figure 7. From this figure, we can see that as

---

[2]We use a commonly used public code `https://github.com/werner-duvaud/muzero-general/tree/continuous`.

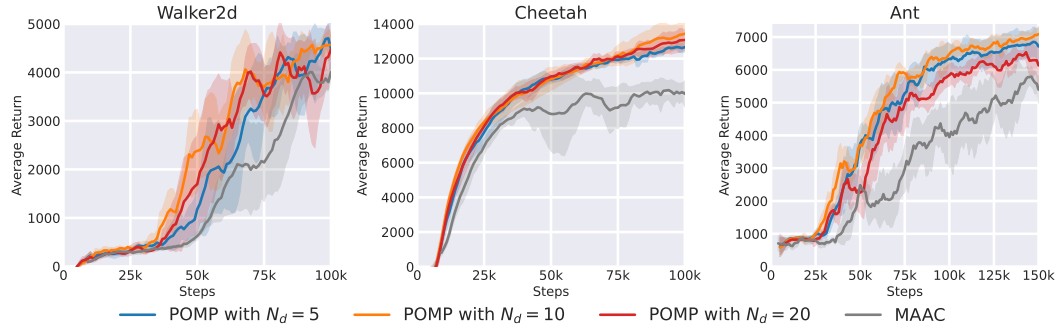

Figure 5: Ablation studies on hyperparameter iteration number $N_d$ in training.

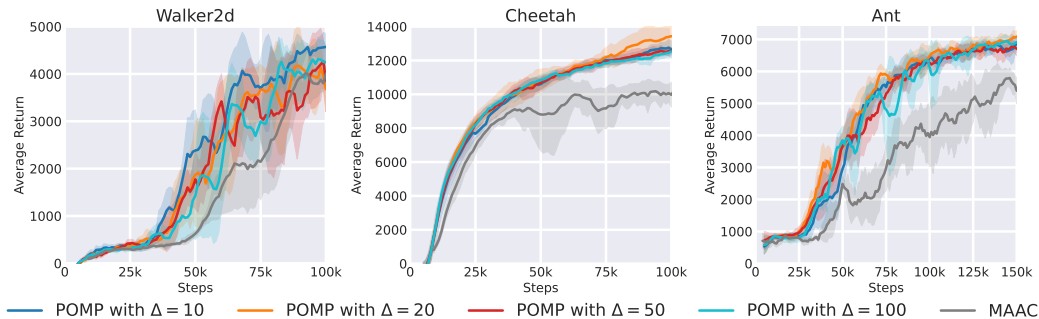

Figure 6: Ablation studies on hyperparameter $\Delta$.

the dimension increases, the gap between our method and the continuous MuZero variant is more and more obvious, which shows the advantage of our method. This also implies that employing Muzero in continuous domain effectively is non-trivial. Since UCT is a principled way to do the exploration in the discrete domain, combining it with our D3P planner for continuous domain will be an interesting research direction in the future.

| Hyper-parameter | Values |
|---|---|
| $\alpha$ in Progressive widening | $\{0.3, 0.4, 0.5, 0.6, 0.7, 0.8\}$ |
| $c_1$ in UCB | $\{1.0, 1.25, 1.5, 2.0\}$ |
| Simulation step $l$ in MCTS | $\{64, 128, 256, 512\}$ |

Table 2: We grid search several important hyper-parameters for the continuous MuZero variant.

### D.3 STUDIES ON THE PLANNING HORIZON

We fix the planning horizon $H$ to be the same as those in MAAC Clavera et al. (2019), since they have systematically studied this hyper-parameter in Section 5.2 of their paper: the gradient error scales poorly with the horizon, and large horizons are detrimental since it magnifies the error on the models. We also add an ablation study to show how the planning horizon influence the performance of our method in Figure 8. The results are consistent with prior work Janner et al. (2019b); Clavera et al. (2019).

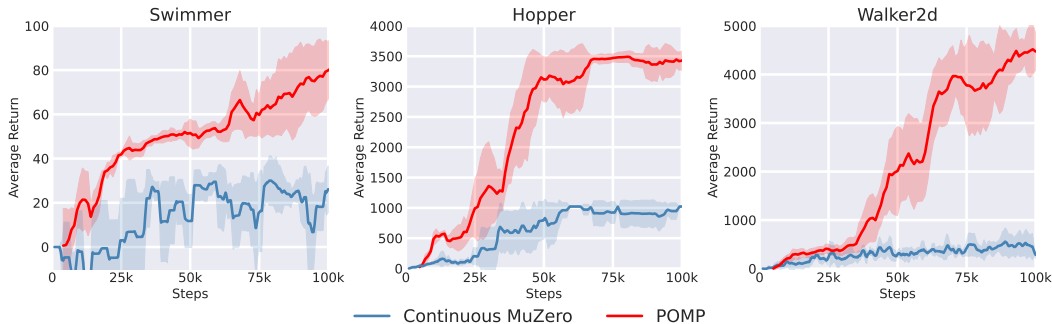

Figure 7: The comparison of a continuous MuZero variant with our method. The dimension of action space for Swimmer, Hopper and Walker2d are 2, 3, and 4, respectively. We can see that as the dimension increases, the gap between of our method and the continuous MuZero variant are more obvious, which shown the advantage of our method.

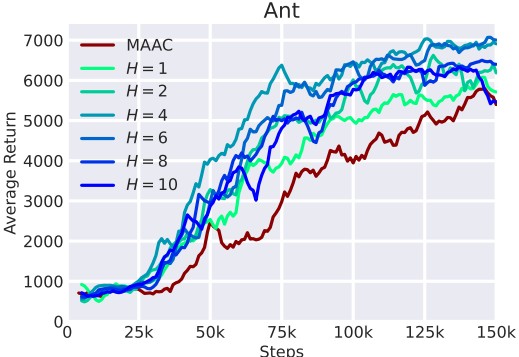

Figure 8: The studies on planning horizon $H$.

## D.4 PLOTTING RESULTS OF DIFFERENT RANDOM SEEDS

Since all the RL literature compare different methods by plotting the mean and standard deviation in their paper, we follow the common practice in our paper. Besides, we also provide the individual run curve in Figure 9. Obviously, if we plot individual runs for each method, it will be messy and unclear for visualization.

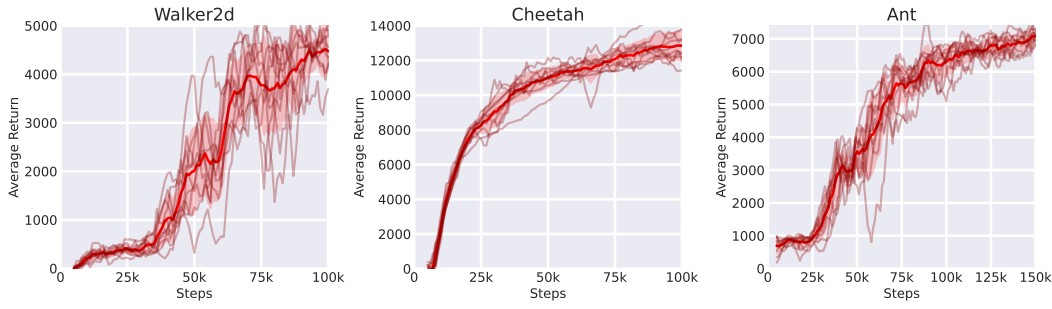

Figure 9: The individual 10 runs of our method.

### D.5 THE IMPACT OF THE NUMBER OF EXPERIMENT RUNS

We have shown the performance of our method with 10 seeds and plotted the mean curve and shaded region with deviation in Figure 1 (the individual 10 runs are also shown in Figure 9). One may still wonder whether the limited number of runs would influence the experimental results. Thus, we run each task with another 20 more seeds (30 seeds, totally), and show the results in Figure 11. Comparing the results of 30 seeds with the results of 10 seeds (shown in Figure 10), we can see that the impact of the number of experiment runs is limited to our method, which does not alter our experimental conclusion. Last, as the RL committee always shows the results with the mean and deviation values, we acknowledge that more runs of each task are needed to show robust and consistent experimental results for RL algorithms.

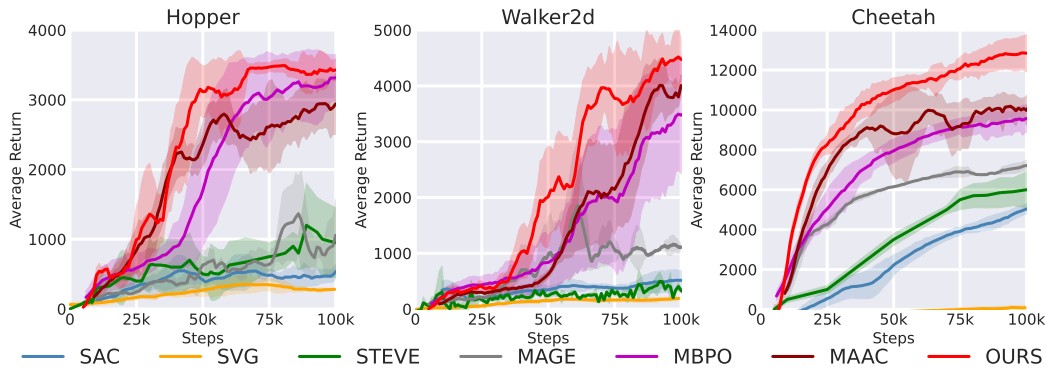

Figure 10: The experimental results with 10 seeds of our method.

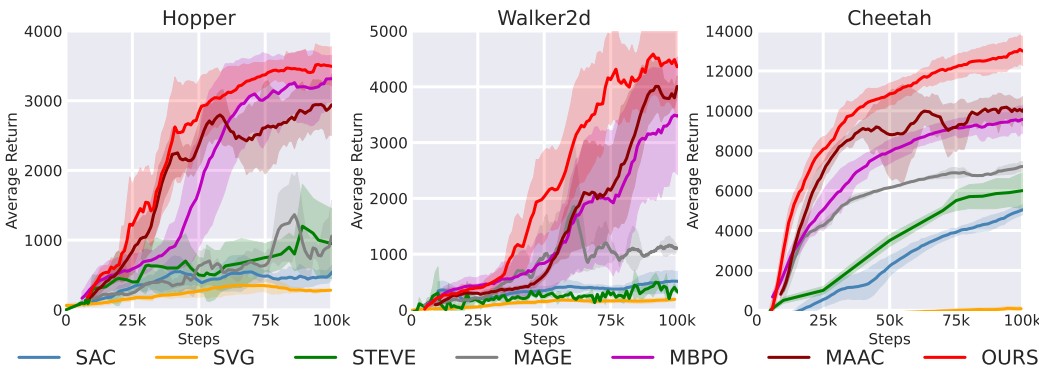

Figure 11: The experimental results with 30 seeds of our method.

## E VECTOR FORM OF OUR D3P PLANNER

For brevity and clear clarification, we treat the action and state as one-dimensional scalars in our main paper. Here we provide the vector form of the derivation of the D3P algorithm.

Consider the state and action are both multi-dimensional vector with dimension $d_{\mathbf{x}}$ and $d_{\mathbf{a}}$. The transition function is now a mapping: $R^{d_{\mathbf{x}}+d_{\mathbf{a}}} \rightarrow R^{d_{\mathbf{x}}}$, the reward function is now a mapping: $R^{d_{\mathbf{x}}+d_{\mathbf{a}}} \rightarrow R^1$. In this situation, $f_{\mathbf{a}}$ is the Jacobin matrix of shape $d_{\mathbf{x}} \times d_{\mathbf{a}}$, whose (i, j)-th entry is $f_{\mathbf{a}_{ij}} = \frac{df_i}{d\mathbf{a}_j}$. Similarly $f_{\mathbf{x}}$ is the Jacobin matrix of shape $d_{\mathbf{x}} \times d_{\mathbf{x}}$, whose (i, j)-th entry is $f_{\mathbf{x}_{ij}} = \frac{df_i}{d\mathbf{x}_j}$.

$r_{\mathbf{a}}$ is the Jacobin matrix of shape $1 \times d_{\mathbf{a}}$, whose (1, j)-th entry is $r_{\mathbf{a}_{1j}} = \frac{dr}{d\mathbf{a}_j}$. $r_{\mathbf{x}}$ is the Jacobin matrix of shape $1 \times d_{\mathbf{x}}$, whose (1, j)-th entry is $r_{\mathbf{x}_{1j}} = \frac{dr}{d\mathbf{x}_j}$.

The objective function of our D3P planner is

$$V(\mathbf{x}, h) = \max_{\mathbf{a}_h}[r(\mathbf{x}_h, \mathbf{a}_h) + V(f(\mathbf{x}_h, \mathbf{a}_h), h+1)]. \tag{11}$$

Denote $Q(\mathbf{x}_h, \mathbf{a}_h) = r(\mathbf{x}_h, \mathbf{a}_h) + V(f(\mathbf{x}_h, \mathbf{a}_h), h+1)$, our goal can be re-expressed as

$$\delta\mathbf{a}_h = \arg\max_{\delta\mathbf{a}} \left[ Q(\mathbf{x}_h, \mathbf{a}_h + \delta\mathbf{a}) \right]. \tag{12}$$

We seek a surrogate objective function $D(\mathbf{x}, \mathbf{a}) \triangleq (Q(\mathbf{x}, \mathbf{a}) - V_{max})^2$, and we then apply first-order Taylor expansion for the Q function $Q(\mathbf{x}, \mathbf{a})$ in $D(\mathbf{x}, \mathbf{a})$,

$$\tilde{D}(\mathbf{x}, \mathbf{a} + \delta\mathbf{a}) = (Q(\mathbf{x}, \mathbf{a}) + Q_{\mathbf{a}}(\mathbf{x}, \mathbf{a})\delta\mathbf{a} - V_{max})^2. \tag{13}$$

So, the optimal action update is $\delta\mathbf{a}^* = -(Q(\mathbf{x}, \mathbf{a}) - V_{max})(Q_{\mathbf{a}}^\top(\mathbf{x}, \mathbf{a})Q_{\mathbf{a}}(\mathbf{x}, \mathbf{a}))^{-1}Q_{\mathbf{a}}^\top(\mathbf{x}, \mathbf{a})$.

Then we introduce a feedback term $\delta\mathbf{x}$, denote

$$\begin{aligned} \mathbf{k} &= (Q(\mathbf{x}, \mathbf{a}) - V_{max})(Q_{\mathbf{a}}^\top(\mathbf{x}, \mathbf{a})Q_{\mathbf{a}}(\mathbf{x}, \mathbf{a}))^{-1}Q_{\mathbf{a}}^\top(\mathbf{x}, \mathbf{a}); \\ \mathbf{K} &= (Q_{\mathbf{a}}^\top(\mathbf{x}, \mathbf{a})Q_{\mathbf{a}}(\mathbf{x}, \mathbf{a}))^{-1}Q_{\mathbf{a}}^\top(\mathbf{x}, \mathbf{a})Q_{\mathbf{x}}(\mathbf{x}, \mathbf{a}), \end{aligned} \tag{14}$$

where the shape of $k$ is $d_a \times 1$ and the shape of $K$ is $d_a \times d_x$. The update rule of the action is given by:

$$\delta\mathbf{a}_h^* = -\mathbf{k} - \mathbf{K}\delta\mathbf{x}. \tag{15}$$

The update rule of $Q_{\mathbf{a}}(\mathbf{x}_h, \mathbf{a}_h)$ and $Q_{\mathbf{x}}(\mathbf{x}_h, \mathbf{a}_h)$ is

$$\begin{aligned} Q_{\mathbf{a}}(\mathbf{x}_h, \mathbf{a}_h) &= r_{\mathbf{a}}(\mathbf{x}_h, \mathbf{a}_h) + V_{\mathbf{x}}(f(\mathbf{x}_h, \mathbf{a}_h), h+1) \cdot f_{\mathbf{a}}(\mathbf{x}_h, a_h); \\ Q_{\mathbf{x}}(\mathbf{x}_h, \mathbf{a}_h) &= r_{\mathbf{x}}(\mathbf{x}_h, \mathbf{a}_h) + V_{\mathbf{x}}(f(\mathbf{x}_h, \mathbf{a}_h), h+1) \cdot f_{\mathbf{x}}(\mathbf{x}_h, \mathbf{a}_h), \end{aligned} \tag{16}$$

and we can calculate $V_{\mathbf{x}}$ by

$$V_{\mathbf{x}} = Q_{\mathbf{x}}(\mathbf{x}_h, \mathbf{a}_h) - Q_{\mathbf{a}}(\mathbf{x}_h, \mathbf{a}_h)\mathbf{K}_h. \tag{17}$$

## F  PROOF OF THEOREM

In this section, we present the proof of the Theorem 1 and Corollary 1 in Section 4. First of all, we summarize the necessary assumptions here.

**Assumption 1.** *The transition $f(x, a)$ and reward function $r(x, a)$ are both continuous and with continuous first and second order derivative. The first and second order derivative are bounded by $L_1$ and $L_2$ respectively.*

$$\|f_x\| + \|f_a\| + \|r_x\| + \|r_a\| \le L_1 \tag{18}$$

$$\|f_{xx}\| + \|f_{xa}\| + \|f_{aa}\| + \|r_{xx}\| + \|r_{xa}\| + \|r_{aa}\| \le L_2 \tag{19}$$

**Assumption 2.** *The variables $Q_a$ calculated in the iteration of D3P are always non-zero.*

### F.1  PROOF OF THE THEOREM 1

Overall speaking, we will use the mathematical induction method to prove the theorem. We will first prove the convergence rate given the trajectory length $H = 2$. Then, we assume the theorem is true in trajectory with length $H = l$, and prove it still holds in trajectory with length $H = l + 1$.

In the proof, we denote the trajectory length as $H$, and denote the location in the trajectory using $h$ where $h \in \{1, 2, \cdots, H\}$. We denote the action in $h$ after update as $a_h'$ where $a_h' = a_h + \delta a_h$. We denote the optimal action as $a_h^*$ where $a_h^* = \arg\max_{a_h} r(x_h^*, a_h) + V(f(x_h^*, a_h), h+1)$ where $x_{h+1}^* = f(x_h^*, a_h^*)$ and $x_1^* = x_1$.

In the proof, we will use $A$ with subscript like $A_1$ to denote some formulation for simplicity and we will give its expression in before we use it. We will use $B$ with subscript like $B_1$ to denote the term related to the error due to using the first-order derivative to approximate the second order derivative. We will use $C$ with subscript like $C_1$ to denote the general constant.

Before the proof, we first recall the update rule of the D3P planner.

$$\delta a_h = -k_h - K_h \delta x_h = -\frac{Q(x_h, a_h) - V_{max}}{Q_a(x_h, a_h)} - \frac{Q_x(x_h, a_h)}{Q_a(x_h, a_h)} \delta x_h, \tag{20}$$

$$Q_a(x_h, a_h) = r_a(x_h, a_h) + V_x(x_{h+1}, h+1) f_a(x_h, a_h), \tag{21}$$

$$Q_x(x_h, a_h) = r_x(x_h, a_h) + V_x(x_{h+1}, h+1) f_x(x_h, a_h), \tag{22}$$

$$V_x(x_h, a_h) = Q_x(x_h, a_h) - Q_a(x_h, a_h) K = Q_x(x_h, a_h) - Q_a(x_h, a_h) \frac{Q_x(x_h, a_h)}{Q_a(x_h, a_h)}. \tag{23}$$

First of all, we consider the case when trajectory length H=2. We calculate the error of $a_1'$ and $a_2'$ in terms of its

$$a_2' - a_2^* = a_2 - a_2^* + \delta a_2 \tag{24}$$

$$= a_2 - a_2^* - \frac{Q(x_2, a_2) - V_{max}}{Q_a(x_2, a_2)} - \frac{Q_x(x_2, a_2)}{Q_a(x_2, a_2)} \delta x \tag{25}$$

$$= \frac{1}{Q_a^2(x_2, a_2)} \left[ Q_a^2(x_2, a_2)(a_2 - a_2^*) - Q_a(x_a, a_2) \left( Q(x_2, a_2) - V_{max} \right) - Q_a(x_2, a_2) Q_x(x_2, a_2) \delta x_2 \right]. \tag{26}$$

Denote $D(x_2, a_2) = \frac{1}{2}(Q(x_2, a_2) - V_{max})^2$. Given the $H = 2$, we have $Q(x_2, a_2) = r(x_2, a_2)$. Therefore, we have $Q_a(x_a, a_2) \left( Q(x_2, a_2) - V_{max} \right) = D_a(x_2, a_2)$. Also, $Q_a(x_2^*, a_2^*) = 0$, according to the definition of $a_h^*$,

By using lemma 1, we have that

$$D_a(x_2, a_2) = D_a(x_2, a_2) - D_a(x_2^*, a_2^*) \tag{27}$$

$$= \int_0^1 D_{aa}(x_2, a_2^* - t(a_2^* - a_2))(a_2 - a_2^*) + D_{ax}(x_2^* - t(x_2^* - x_2), x_2)(x_2 - x_2^*) dt. \tag{28}$$

Denote $A_1 = \int_0^1 D_{aa}(x_2, a_2^* - t(a_2^* - a_2))(a_2 - a_2^*) dt$ and $A_2 = \int_0^1 D_{ax}(x_2^* - t(x_2^* - x_2), x_2)(x_2 - x_2^*) dt$ and consider the first and second term in equation 26, we have

$$Q_a^2(x_2, a_2)(a_2 - a_2^*) - Q_a(x_a, a_2) \left( Q(x_2, a_2) - V_{max} \right) \tag{29}$$

$$= Q_a^2(x_2, a_2)(a_2 - a_2^*) - D_a(x_2, a_2) \tag{30}$$

$$= Q_a^2(x_2, a_2)(a_2 - a_2^*) - A_1 - A_2. \tag{31}$$

We first consider the $A_1$ term. Denote $h_1(x, a) = Q_a^2(x, a) - D_{aa}(x, a)$. Denote $B_1 = \int_0^1 h_1(x_2, a_2^* - t(a_2^* - a_2)) dt$.

$$Q_a^2(x_2, a_2)(a_2 - a_2^*) - A_1 \tag{32}$$

$$= (a_2 - a_2^*) \left[ Q_a^2(x_2, a_2) - \int_0^1 D_{aa}(x_2, a_2^* - t(a_2^* - a_2))dt \right] \tag{33}$$

$$= (a_2 - a_2^*) \left[ \int_0^1 Q_a^2(x_2, a_2) - D_{aa}(x_2, a_2^* - t(a_2^* - a_2))dt \right] \tag{34}$$

$$\leq \|a_2 - a_2^*\| \left\| \int_0^1 Q_a^2(x_2, a_2) - D_{aa}(x_2, a_2^* - t(a_2^* - a_2))dt \right\| \tag{35}$$

$$\leq \|a_2 - a_2^*\| \left[ \int_0^1 Q_a^2(x_2, a_2) - Q_a^2(x_2, a_2^* - t(a_2^* - a_2)) + Q_a^2(x_2, a_2^* - t(a_2^* - a_2)) - D_{aa}(x_2, a_2^* - t(a_2^* - a_2))dt \right] \tag{36}$$

$$\leq \|a_2 - a_2^*\| \left[ \int_0^1 L_2(1-t)(a_2 - a_2^*)dt + \int_0^1 h_1(x_2, a_2^* - t(a_2^* - a_2))dt \right] \tag{37}$$

$$\leq \|a_2 - a_2^*\|^2 \frac{L_2}{2} + B_1 \|a_2 - a_2^*\|. \tag{38}$$

Now, we will consider the $A_2$ term and the third term in equation 26. Denote $h_2(x, a) = D_{ax}(x, a) - Q_a(x, a)Q_x(x, a)$. Denote $B_2 = \int_0^1 h_2(x_2^* - t(x_2^* - x_2), a_2)dt$.

$$- A_2 - Q_a(x_2, a_2)Q_x(x_2, a_2)\delta x_2 \tag{39}$$

$$= - \int_0^1 D_{ax}(x_2^* - t(x_2^* - x_2), a_2)(x_2 - x_2^*)dt - Q_a(x_2, a_2)Q_x(x_2, a_2)\delta x_2 \tag{40}$$

$$= - \int_0^1 D_{ax}(x_2^* - t(x_2^* - x_2), a_2)(x_2 - x_2^*)dt - Q_a(x_2, a_2)Q_x(x_2, a_2)(x_2' - x_2) \tag{41}$$

$$= - \int_0^1 Q_a(x_2^* - t(x_2^* - x_2), a_2)Q_x(x_2^* - t(x_2^* - x_2), a_2)(x_2 - x_2^*)dt \tag{42}$$

$$- Q_a(x_2, a_2)Q_x(x_2, a_2)(x_2' - x_2) + \int_0^1 h_2(x_2^* - t(x_2^* - x_2), a_2)(x_2 - x_2^*)dt \tag{43}$$

$$= - Q_a(x_2, a_2)Q_x(x_2, a_2)(x_2 - x_2^*) - Q_a(x_2, a_2)Q_x(x_2, a_2)(x_2' - x_2) \tag{44}$$

$$+ \frac{L_2^2 L_1}{2}(x_2 - x_2^*)^2 + \int_0^1 h_2(x_2^* - t(x_2^* - x_2), a_2)(x_2 - x_2^*)dt \tag{45}$$

$$= Q_a(x_2, a_2)Q_x(x_2, a_2)(x_2^* - x_2') + \frac{L_2^2 L_1}{2}(x_2 - x_2^*)^2 + \int_0^1 h_2(x_2^* - t(x_2^* - x_2), a_2)(x_2 - x_2^*)dt \tag{46}$$

$$\leq L_1^3 \|a_1' - a_1^*\| + \frac{L_2^2 L_1^3}{2} \|a_1 - a_1^*\|^2 + B_2 \|a_1 - a_1^*\|. \tag{47}$$

Summarize the conclusion, we can prove now

$$a_2' - a_2^* \leq \mathcal{O}\left( (a_2 - a_2^*)^2 + (a_2 - a_2^*) + (a_1 - a_1^*)^2 + (a_1' - a_1^*) \right). \tag{48}$$

The next task is to prove the $a_1' - a_1^* \leq \mathcal{O}\left( (a_1' - a_1^*)^2 + (a_2' - a_2^*)^2 + (a_1' - a_1^*) + (a_2' - a_2^*) \right)$. Similarly, we have

$$a_1' - a_1^* = a_1 + \delta a_1 - a_1^* \tag{49}$$

$$= a_1 - a_1^* + \frac{Q(x_1, a_1) - V_{max}}{Q_a(x_1, a_1)} \tag{50}$$

$$= \frac{1}{Q_a^2(x_1, a_1)} \left[ Q_a^2(x_1, a_1)(a_1 - a_1^*) + Q_a(x_1, a_1)(Q(x_1, a_1) - V_{max}) \right]. \tag{51}$$

We have $Q_a(x_1, a_1^*) = 0$ according to the definition of $a_1^*$.

Using Lemma 1, denote $B_3 = \int_0^1 h_1(x_1, a_1^* - t(a_1^* - a_1))dt$. we have

$$Q_a^2(x_1, a_1)(a_1 - a_1^*) + Q_a(x_1, a_1)(Q(x_1, a_1) - V_{max}) \tag{52}$$

$$= (a_1 - a_1^*) \int_0^1 Q_a^2(x_1, a_1) - D_{aa}(x_1, a_1^* - t(a_1^* - a_1))dt \tag{53}$$

$$\leq \|a_1 - a_1^*\|^2 \frac{L_2}{2} + B_3\|a_1 - a_1^*\|. \tag{54}$$

Thus, we have

$$\|a_1' - a_1^*\| \leq \frac{1}{Q_a^2(x_1, a_1)} \left[ \frac{L_2}{2}\|a_1 - a_1^*\|^2 + B_3\|a_1 - a_1^*\| \right] \tag{55}$$

$$\|a_2' - a_2^*\| \leq \frac{1}{Q_a^2(x_2, a_2)} \left[ \frac{L_2}{2}\|a_2 - a_2^*\|^2 + B_1\|a_2 - a_2^*\| \right. \tag{56}$$

$$\left. + \left( \frac{L_2 L_1^3}{2Q_a^2(x_1, a_1)} + B_1 L_1 \right) \|a_1 - a_1^*\|^2 + \left( \frac{B_3 L_1^3}{Q_a^2(x_1, a_1)} + B_2 \right) \|a_1 - a_1^*\| \right]. \tag{57}$$

For simplicity, we can write

$$\|a_h' - a_h^*\| \leq C(\|a_1 - a_1^*\|^2 + \|a_2 - a_2^*\|^2) + B(\|a_1 - a_1^*\|^2 + \|a_2 - a_2^*\|) \quad \text{for } h = 1, 2. \tag{58}$$

Up to here, we prove the theorem is true in trajectory with horizon $H = 2$.

Now, using induction method, suppose the theorem is true for $H = l - 1$. The induction hypothesis means that for the following problem (denote as $P(l-1)$), there exist two constant $C$ and $B$, such that for $\forall h \in \{1, 2, \cdots l - 1\}$, we have $\|a_h' - a_h^*\| \leq C \sum_{k=1}^{l-1} \|a_h - a_h^*\|^2 + B \sum_{k=1}^{l-1} \|a_h - a_h^*\|$.

$$\min_{a_1, \cdots, a_{l-1}} \sum_{k=1}^{l-1} r(x_k, a_k) \tag{59}$$

$$x_{k+1} = f(x_k, a_k) \quad k = 1, \cdots, l - 2. \tag{60}$$

What we need to prove is for the new problem with $H = l$ (denote as $p(l)$), the theorem still holds.

The main idea is to merge the reward function in last two timesteps into one, and then prove the $\delta a_{l-1}$ is the same as the one in the problem $P(l-1)$ which is denoted as $\hat{\delta} a_{l-1}$. Then, according to the update rule, for $h < l - 1$, the $\delta a_h = \hat{\delta} a_h$ also holds. For $h = l$, the theorem can be proved using the exact the same process as we prove $a_2' - a_2^*$ in $H = 2$. Combining all these conclusions, we can then prove the theorem holds for the problem $p(l)$ and thus the proof finished.

Here we show how can we construct a new reward function by merge two reward function. Denote $R(x_{l-1}, a_{l-1}) = r(x_{l-1}, a_{l-1}) + r(x_l, a_l - \frac{Q(x_h, a_h) - V_{max}}{Q_a(x_l, a_l)} - \frac{Q_x(x_l, a_l)}{Q_a(x_l, a_l)}(x_l' - x_l))$ where $x_l' = f(x_{l-1}, a_{l-1})$.

In new problem, the update for action

$$\delta a_{l-1} = -k_{l-1} - K_{l-1}\delta x_{l-1}. \tag{61}$$

$$k_{l-1} = \frac{Q(x_{l-1} - V_{max})}{Q_a(x_{l-1}, a_{l-1})} \tag{62}$$

$$K_{l-1} = \frac{Q_x(x_{l-1}, a_{l-1})}{Q_a(x_{l-1}, a_{l-1})}. \tag{63}$$

$$Q_a(x_{l-1}, a_{l-1})) = R_a(x_{l-1}, a_{l-1}) + V_x(x_l, l) f_a(x_{l-1}, a_{l-1}) \tag{64}$$

$$= r_a(x_{l-1}, a_{l-1}) - r_a(x_l, a_l) \frac{Q_x(x_l, a_l)}{Q_a(x_l, a_l)} f_a(x_{l-1}, a_{l-1}) + V_x(x_l, l) f_a(x_{l-1}, a_{l-1}) \tag{65}$$

$$= r_a(x_{l-1}, a_{l-1}) - r_a(x_l, a_l) \frac{Q_x(x_l, a_l)}{Q_a(x_l, a_l)} f_a(x_{l-1}, a_{l-1}) + r_x(x_l, a_l) f_a(x_{l-1}, a_{l-1}). \tag{66}$$

$$Q_x(x_{l-1}, a_{l-1})) = R_x(x_{l-1}, a_{l-1}) + V_x(x_l, l) f_x(x_{l-1}, a_{l-1}) \tag{67}$$

$$= r_x(x_{l-1}, a_{l-1}) - r_a(x_l, a_l) \frac{Q_x(x_l, a_l)}{Q_a(x_l, a_l)} f_x(x_{l-1}, a_{l-1}) + V_x(x_l, l) f_x(x_{l-1}, a_{l-1}) \tag{68}$$

$$= r_x(x_{l-1}, a_{l-1}) - r_a(x_l, a_l) \frac{Q_x(x_l, a_l)}{Q_a(x_l, a_l)} f_x(x_{l-1}, a_{l-1}) + r_x(x_l, a_l) f_x(x_{l-1}, a_{l-1}). \tag{69}$$

It can be easily verified, $\delta a_{l-1} = \hat{\delta} a_{l-1}$.

## F.2 PROOF OF THE COROLLARY 1

If we do not consider the feedback term, $K = 0$. The new update rule will be

$$\delta a_h = -k_h = -\frac{Q(x_h, a_h) - V_{max}}{Q_a(x_h, a_h)} \tag{70}$$

$$Q_a(x_h, a_h) = r_a(x_h, a_h) + V_x(x_{h+1}, h+1) f_a(x_h, a_h) \tag{71}$$

$$Q_x(x_h, a_h) = r_x(x_h, a_h) + V_x(x_{h+1}, h+1) f_x(x_h, a_h) \tag{72}$$

$$V_x(x_h, a_h) = Q_x(x_h, a_h). \tag{73}$$

According the proof of Theorem 1, for $\forall h$,

$$a'_h - a_h^* \leq C_1 \sum_{k=1}^{H} \|a_k - a_k^*\|^2 + C_2 \sum_{k=1}^{H} \|a_k - a_k^*\| + \left\| \frac{Q_x(x_h, a_h)}{Q_a(x_h, a_h)} x'_h - x_h \right\| \tag{74}$$

$$\leq C_1 \sum_{k=1}^{H} \|a_k - a_k^*\|^2 + C_2 \sum_{k=1}^{H} \|a_k - a_k^*\| + \left\| \frac{Q_x(x_h, a_h)}{Q_a(x_h, a_h)} \right\| \|x'_h - x_h\|. \tag{75}$$

where $x'_h = f(x'_{h-1}, a_{h-1} + \delta a_{h-1})$, $x_h = f(x_{h-1}, a_{h-1})$. Using Taylor expansion, there exist a constant $C$ such that

$$x'_h - x_h = f_x(x_{h-1}, a_{h-1})(x'_{h-1} - x_{h-1}) + f_a(x_{h-1}, a_{h-1})(\delta a_{h-1}) + C((x'_{h-1} - x_{h-1})^2 + (a'_{h-1} - a_{h-1})^2). \tag{76}$$

If $x'_{h-1} - x_{h-1} \leq 1$, we can ignore the error of the first-order Taylor expansion,

$$x'_h - x_h = \sum_{i=h-1}^{1} \Pi_{j=i+1}^{h-1} f_x(x_j, a_j) \left[ f_a(x_i, a_i) \delta a_i + C(\delta a_i^2) \right]. \tag{77}$$

And the corollary can be proved.

**Lemma 1.** *Denote the function $f(x)$ have continues derivative. Denote the first order derivative of function $f(x)$ as $f_x(x)$. Then we have*

$$f(x_2) - f(x_1) = \int_0^1 f_x(x_1 - t(x_1 - x_2))(x_2 - x_1) dt. \tag{78}$$

*For multi-variable function $f(x, y)$, we have*

$$f(x_2, y_2) - f(x_1, y_1) = \int_0^1 f_x(x_1 - t(x_1 - x_2), y_2)(x_2 - x_1) + f_y(x_2, y_1 - t(y_1 - y_2))(y_2 - y_1)dt. \tag{79}$$

*Proof of the Lemma 1.* We first prove the single-variable version. Denote $g(t) = f(x_1 - t(x_1 - x_2))$, it is easy to verify that

$$f(x_2) - f(x_1) = g(1) - g(0). \tag{80}$$

According to the fundamental theorem of calculus, we have

$$g(1) - g(0) = \int_0^1 \frac{dg(t)}{dt}dt \tag{81}$$

$$= \int_0^1 \frac{df(x_1 - t(x_1 - x_2))}{dt}dt \tag{82}$$

$$= \int_0^1 \frac{df(x_1 - t(x_1 - x_2))}{dx} \frac{d(x_1 - t(x_1 - x_2))}{dt}dt \tag{83}$$

$$= \int_0^1 f_x(x_1 - t(x_1 - x_2))(x_2 - x_1)dt. \tag{84}$$

Then, we prove the multi-variable version. Denote $g(t) = f(x_1 - t(x_1 - x_2), y_1 - t(y_1 - y_2))$.

$$f(x_2, y_2) - f(x_1, y_1) \tag{85}$$

$$= g(1) - g(0) \tag{86}$$

$$= \int_0^1 \frac{dg(t)}{dt}dt \tag{87}$$

$$= \int_0^1 \frac{df(x_1 - t(x_1 - x_2), y_1 - t(y_1 - y_2))}{dt}dt \tag{88}$$

$$= \int_0^1 \frac{df(x_1 - t(x_1 - x_2))}{dx} \frac{d(x_1 - t(x_1 - x_2))}{dt} + \frac{df(x_1 - t(x_1 - x_2))}{dy} \frac{d(y_1 - t(y_1 - y_2))}{dt}dt \tag{89}$$

$$= \int_0^1 f_x(x_1 - t(x_1 - x_2), y_2)(x_2 - x_1) + f_y(x_2, y_1 - t(y_1 - y_2))(y_2 - y_1)dt. \tag{90}$$

$\square$

**Lemma 2.** *Denote $D(x, a) = \frac{1}{2}(Q(x, a) - V_max)^2$, denote $h_1(x, a) = D_{aa}(x, a) - Q_a^2(x, a)$, $h_2(x, a) = D_{ax} - Q_a(x, a)Q_x(x, a)$, we have*

$$h_1 = Q_{aa}(Q(x, a) - V_{max}), \tag{91}$$

$$h_2 = Q_{ax}(Q(x, a) - V_{max}) \tag{92}$$

*Proof of Lemma 2.*

$$D_{ax} = \frac{d^2 D(x, a)}{dadx} = \frac{d^2 \left[\frac{1}{2}(Q(x, a) - V_{max})^2\right]}{dadx} = \frac{d\left[Q_a(x, a)(Q(x, a) - V_{max})\right]}{dx} \tag{93}$$

$$= Q_a(x, a)Q_x(x, a) + Q_{ax}(Q(x, a) - V_{max}) \tag{94}$$

Similarly, we can prove that

$$D_{aa} = Q_a(x, a)Q_a(x, a) + Q_{aa}(Q(x, a) - V_{max}). \tag{95}$$

$\square$

