# OpenReview forum: "Making Better Decision by Directly Planning in Continuous Control"
_ICLR.cc/2023/Conference — ICLR 2023 poster_

### Official Review · Reviewer_Pekh · 2022-10-23

**Confidence:** 4
**Correctness:** 3
**Technical Novelty And Significance:** 3
**Empirical Novelty And Significance:** 3
**Recommendation:** 8

**Clarity, Quality, Novelty And Reproducibility:**

Quality:
The paper presents a novel model-based RL method, with some analysis on the convergence guarantees of one of their components
Clarity
The paper is clearly written and easy to follow
Originality:
The paper present a novel and original method
Reproducibility:
The method is straightforward and easy to implement, the authors provide pseudocode, training and architectural details as well as the source code. Reproducibility should not be an issue even though I did not run the source code myself.

**Strength And Weaknesses:**

Strengths:
- The paper is clearly written and easy to follow
- A theoretical analysis of the proposed method is shown, with relative convergence proof
- An extensive empirical evaluation is performed showing the effectiveness of the method

Weaknesses:
- The authors should better express the limitations of the method. They should clearly state that the method is meant to be used in deterministic environments as this is heavily used in the definition of their planner.
- They should further clarify that the convergence proof applied only to the planner with a fixed (probably wrong) learned transition model and reward, i.e. D3P recovers the optimal action of a wrong model. It should be made clear that no convergence guarantee is given on the whole algorithm, i.e. no guarantees on the performance of the learned policy compared to the optimal policy of the MDP
- The empirical evaluation is thorough but important baselines are missing in my opinion. I would have expected to see a comparison with MuZero combined with Continuous  UCT (Progresstive Widening). This because the model and reward learning procedure of POMP is very similar to the one of MuZero. Moreover, continuous UCT, like D3P is known to converge to the optimal sequence of actions given a fixed environment model. With the current empirical evaluation, It is unclear if POMP would outperform MuZero + Continuous UCT, which is a fairly important baseline for model-based RL.

**Summary Of The Paper:**

The paper proposes a new model-based RL framework for continuous control in deterministic environments. The authors propose Policy Optimization with Model Planning (POMP) an algorithm that uses Differential Dynamic Programming (DDP) as the planner module. A specific instantiation of DDP is introduced which uses (D3P) which uses neural networks as differentiable models and first order taylor approximation to improve the computational efficiency of the planner. The method is shown to converge to the optimal sequence of actions, given a fixed learned model. Extensive experimental evaluations are performed, together with an ablation study that show the effectiveness of the proposed method compared with  state-of-the-art methods

**Summary Of The Review:**

Overall I am positive towards the paper. The author propose a novel original method, that works well in practice and that has convergence guarantees under some mild assumptions. It is unclear though how the proposed method performs compared to MuZero for continuous actions, which is an important baseline for the setting, indeed both methods were cited repeatedly in the paper.
I would increase my score if this comparison is performed, or alternatively if a valid reason for omitting this comparison is given.

---

> ### Author Response · Authors · 2022-11-14
> **Response to Reviewer Pekh**
>
> We sincerely thank the reviewer for recognizing our motivation and novelty. Thank you for your detailed comments and suggestions.  We provide the following responses to your review comments.
>
> > **The limitation of the application scope of D3P planner.**
>
> **A:** We have revised Section 4.1 to clarify our limitations. Please note that although the current presentation of our method can only be applied in the deterministic environment, the D3P planner can be easily extended to the stochastic environment with a reparameterization trick (Such as normal distribution noise in [1]). Actually, in the code implementation, we follow our baseline method to assume the next state is in a normal distribution (to capture the uncertainty, the same as in MAAC and MBPO). The neural network outputs the mean and the variance of the next state. Also, thank you for your interesting questions, investigating efficient planners for arbitrary stochastic transition will be interesting future work.
>
> > **The influence of the learned model error.**
>
> **A:** Yes, you are right. Sorry that you might be misled by our paper writing. Theorem 1 shows that, given a model, the D3P planner can optimize the action sequence towards the optimal action sequence of the given model. According to the paper structure, we discuss the model error and the solution to alleviate its influence in Subsection 4.2. As we stated in the second paragraph in Subsection 4.2: “One key problem that needs to be resolved before applying the D3P planner is how to avoid misleading planning due to the limited generalization ability of the learned model. Such a problem cannot be ignored as long as the ground-truth model is unknown, which can only be learned by data with function approximation.” To combine our D3P planner with a learned model, we design two strategies in Section 4.2 and verify their effectiveness in the experiment section, especially in Figure 4(c).
>
> > **Comparison with MuZero for continuous actions (Progressive Widening).**
>
> **A:** Thank you for your important information. We have added a baseline with the learned model + MCTS + Continuous UCT (Progressive Widening). Our implementation is based on [muzero-general](https://github.com/werner-duvaud/muzero-general/tree/continuous).  We grid search 3 important hyperparameters for the new baseline and report the best results. The results are reported in Appendix D.2. From the experiments, we can see that as the dimension increases, the gap between our method and the continuous MuZero variant is more obvious, which clearly shows the advantage of our method. This also implies that employing Muzero in continuous domain effectively is non-trivial. Since UCT is a principled way to do the exploration in the discrete domain, combining it with our D3P planner for continuous domain will be an interesting research direction in the future.
>
> [1]. Kingma, Diederik P., and Max Welling. "Auto-encoding variational bayes." arXiv preprint arXiv:1312.6114 (2013).

---

> > ### Comment · Reviewer_Pekh · 2022-11-19
> > **Response to authors**
> >
> > I thank the authors for addressing my concerns as well as the concerns of the other reviewers. The paper in its current form is a stronger submission compared to the original. That is why I will gladly raise my score to an accept.

---

> > > ### Author Response · Authors · 2022-11-21
> > > **Thanks!**
> > >
> > > We are very glad that our responses can address your concerns. Thank you for your recognition of the idea of our work and your efforts on improving our work!

---

### Official Review · Reviewer_GhwF · 2022-10-24

**Confidence:** 4
**Clarity, Quality, Novelty And Reproducibility:** See above.
**Correctness:** 3
**Technical Novelty And Significance:** 3
**Empirical Novelty And Significance:** 3
**Recommendation:** 8

**Strength And Weaknesses:**

Thanks for this enjoyable review. The paper is very very well written. One can easily follow the paper and understand the bigger picture. Such a clear presentation of the ideas and results is very rare. Furthermore, the experimental evaluation is great. The paper first shows the performance in terms of rewards and conducts ablation studies answering specific questions. From a research project execution point of view, the paper is an A+. The research idea/question is ok but not super innovative, which is fine. Therefore, the paper is a very good contribution to ICLR. However, there are also quite some shortcomings, which prevent this paper from being an excellent paper instead of "only" a good paper.

The paper unfortunately misses a large part of the model-based RL literature. Many papers have already used an MPC approach that optimizes an action sequence WITH A LEARNED MODEL online to obtain better actions. There are initial versions that do not rely on a policy, e.g. Pets (Chua et. al.), PDDM (Nagabandi et. al), iCEM (Pinneri et. al.), and later versions that distill the samples into a policy using any deep RL algorithm and use the learned policy as a prior e.g., EVALUATING MODEL-BASED PLANNING AND PLANNER AMORTIZATION FOR CONTINUOUS CONTROL (Byravan et. al.). These are just a few exemplary papers and there are 10-30 papers on MPC with learned models from various groups. Somehow, this paper forgets to address this line of research.

One interesting point of all of these approaches is that they rely on zero-order optimization to solve the local optimization problem. A frequently discussed question was; Why is everybody using in-efficient zero-order optimization methods? Wouldn't gradient-based approaches be more efficient? One common hypothesis was, that deep networks are good for forward prediction but the gradients are not great. Therefore, it is believed the backprop through time literature never took off in deep RL. However, this paper uses a gradient-based optimization to solve the MPC problem. It also performs an ablation study of the locally quadratic problem to the naive SGD backprop to time. So it is very close to giving more insights into this question. Therefore, it would be great to perform more ablation studies on whether the locally quadratic approach performs better than the zero-order online optimization with learned models and include this line of research within the paper.

Further minor points:

* `In most of the model-based RL algorithms, the learned predictive models always play an auxiliary role to only affect the decision-making by helping the policy learning.`
Given that there are many papers doing MPC with learned models/rewards, this statement is wrong.

 * `differential dynamic programming (DDP) (DE O. PANTOJA, 1988; Tassa et al., 2012) algorithm` The capitalization of the citation is wrong.

*  `optimal bellman equation`  Bellman has to be capitalized.

* Equation 4 and the following are sloppy. The equations treat vectors as scalars and easily divide through vectors, i.e., Q_a, Q_x. This section should be rewritten to properly treat vectors and remove the ambiguity of what these equations mean.

* The locally quadratic DDP optimization looks very close to identical to iLQR from Tassa et. al.. The paper needs to add either a section to highlight the differences to the iLQR optimization approach. When the only difference is the learned deep dynamics models, the paper needs to explicitly state: "We use iLQR with learned models for MPC" and not slap a new name on an existing technique, i.e., Deep Differential Dynamic Programming (D3P).

*  `Please note that we only use this conservative term during evaluation, as we want to encourage exploration when training`
Does this statement implicitly mean, if we include the conservative term in the exploration it is too little exploration and our learning curve is below the baselines? If you use the greedy algorithm in your evaluation to plot the learning curve, do you also use the deterministic policies of your baselines during the evaluation?

* The MBPO state augmentation to push sample efficiency is never really mentioned in the main paper but only shown in algorithm 2 in the appendix. It would be great to make this more explicit in the main paper. Furthermore, it would be great to perform an ablation study without this sample augmentation to see how the pure DDP MPC increases sample efficiency.

**Summary Of The Paper:**

The paper proposes to use a planner solving a locally quadratic program to improve the action selection during the rollouts. The better action selection should improve the learning performance and enable the agent to achieve a higher reward with fewer samples. The experiments show this improved sample efficiency and the agent learns faster to obtain higher rewards.


**Summary Of The Review:**

The execution and presentation of this research project are very good. One can follow the motivation, approach, and results. Very few papers achieve this clarity. However, the paper is also very far from perfect. Missing out on the related work on zero-order MPC with learned models is quite bad. Not clearly stating the differences between the iLQR optimization and the sloppy vector math significantly reduces the quality of the paper. Further ablation study comparing the locally quadratic online optimization to zero order optimization would be great to come closer to answering gradient based vs sample-based MPC with learned models question. Therefore, the details of the work can be significantly improved and make this paper from okayish to excellent.

---

> ### Author Response · Authors · 2022-11-14
> **Response to Reviewer GhwF [Part 1]**
>
> Thank you very much for your positive comments on our work. As you are saying, we are trying our best to make our presentation clearer, and we sincerely thank you for your detailed comments which help us significantly improve the quality of the paper. We provide our response to your concerns below.
> > **Concerns about the related work.**
>
> **A**: Thanks for your questions. We provide the following answers.
> - Thank you for pointing out the line of the research. We have carefully read these papers and added them to the related work section in the main paper. More discussions about these works are in Appendix A.
> - As you mentioned, the zero-order method will suffer from the curse of dimension problem. We have added two baselines about the random shooting method and the cross-entropy method in Figure 3 to verify the intuition. The gaps between our method and the CEM planner (denoted by CEM) and the random-shooting planner (denoted by Random-shooting) clearly show the advantage of our gradient-based method.
> - The usage of the first-order method is not as trivial as the zero-order method. To use the first-order method, we need a differentiable environment, which is a stronger requirement for a zero-order simulator. Learning a model is a feasible way to get an approximated differentiable environment, but we need to carefully handle the model error since we do not have the ground truth model. There are indeed some gradient-based methods in Model-based DRL for policy learning such as MAAC, but they are not efficient enough.
> - To answer the question about the necessity of using the D3P planner rather than other first-order methods (such as first-order policy learning and SGD-based planner), we have conducted the experiment and shown the results in Figure 3. As shown in Figure 3, we use ablation experiments to show two reasons why previous gradient-based methods are not efficient. First, only using first-order information to do policy learning (i.e., MAAC) is not efficient for decision-making. Planning is crucial for efficiency. Second, the SGD-based planner is not efficient compared to our D3P planner. Our theoretical results partially explain why this happens: D3P considers the feedback information, and it would have a near quadratic convergence rate if the second derivative of the transition function is small.
>
> > **Concerns about the correctness of the statement in the introduction.**
>
> **A**: Thanks for your constructive comments. We have revised our introduction, especially the second paragraph, to make it clear. Now, we state: “There are mainly two directions to leverage learned predictive model in model-based RL. In the first class, the predictive models always play an auxiliary role to affect decision-making by only helping policy learning [1][2]. In the second class, we sample pathwise trajectories and then score these sampled actions [3]. Our work falls into the second class, as directly using the model as a planner (rather than only helping the policy learning) to achieve this goal might be a more straightforward approach when the real/learned environment model is at hand.
>
> > **About typos.**
>
> **A**: Thanks for your careful review. We have fixed the typos in the updated version.
>
> > **Concerns about the Equation 4 and the following.**
>
> - Thanks for your suggestions, we have added several explanations for the meaning of these equations' notation to make it clearer. Please note that these formulations are conceptually correct if the state and action are with 1-dimension as we state at the beginning of this subsection (third paragraph in Subsection 4.1). We use the “Notions” paragraph in the “Preliminary” (Section 3) to explain the definition of $f_x, f_a, r_x, r_a, Q_x, Q_a$.
> - We put the vector form of the derivation of the D3P planner in Appendix E and put a reference in the main paper (In Section 4.1, above equation 2).
> - Since it is trivial to derive the D3P planner from one dimension case to the multiple-dimension case, we then use the scalar form in the main paper. We think the scalar form is clear and easy to follow for those who are not familiar with the derivation.

---

> > ### Author Response · Authors · 2022-11-14
> > **Response to Reviewer GhwF [Part 2]**
> >
> > > **The difference between our D3P planner and iLQR.**
> >
> > **A**: Both iLQR and our D3P planner are motivated by DDP method, they look similar naturally. But we should clarify that our method has several differences compared with iLQR, and these differences are well-designed to incorporate the neural network model.   (1) Computing the second-order derivative of the neural network based model will be computationally costly (Hessian matrix). In our method, we only rely on the first-order derivative of the model. (2) The iLQR method uses the second-order Talyor expansion of the Q-value function to handle the local optimization problem. But it is hard to guarantee that the hessian matrix is a negative definite matrix, which is a necessary condition for convergence. Here, we construct an auxiliary target function $D$ and use a first-order Taylor expansion for the Q function inside of the $D$ function to guarantee the non-positive definite matrix.  We have revised our related work section and made a detailed discussion about this content.
> >
> > > **Two questions about the conservative term during the evaluation.**
> >
> > Thanks for your questions. We provide the following answers.
> > - “If we include the conservative term in the exploration, it is too little exploration, and our learning curve is below the baselines?” Yes, you are right. Actually, we are planning to investigate how to incorporate the exploration strategy into our planner during the training phase in the near future. The strategy like UCT may be an interesting direction for future work but this is not the focus of this paper.
> > - “Do you also use the deterministic policies of your baselines during the evaluation”. Yes. During the evaluation phase, the policy of baseline works is all fixed as a deterministic policy as done in their original work.
> >
> > > **Question about state augmentation.**
> >
> > **A**: Please note that our planner is built upon the framework of MBPO [1] and MAAC [2]. Therefore, the sample efficiency of our method is compared with MBPO and MAAC which also use the same state augmentation strategy. So, the improvement of the sample efficiency is not relevant to the state augmentation strategy. The comparison is fair. We have added a citation in Section 5 in the main paper and have added an explanation in Appendix C.3.
> >
> > [1].	Janner, Michael, Justin Fu, Marvin Zhang, and Sergey Levine. "When to trust your model: Model-based policy optimization." Advances in Neural Information Processing Systems 32 (2019).
> >
> > [2].	Clavera, Ignasi, Yao Fu, and Pieter Abbeel. "Model-Augmented Actor-Critic: Backpropagating through Paths." In International Conference on Learning Representations. 2019.
> >
> > [3].	Schrittwieser, Julian, Ioannis Antonoglou, Thomas Hubert, Karen Simonyan, Laurent Sifre, Simon Schmitt, Arthur Guez et al. "Mastering atari, go, chess and shogi by planning with a learned model." Nature 588, no. 7839 (2020): 604-609.

---

> > > ### Comment · Reviewer_GhwF · 2022-11-18
> > > **Post-Rebuttal Comment**
> > >
> > > The authors did a nice update of the paper and a thorough rebuttal. Now the paper meets the expectations for an ICLR paper. I adapted my score to accept.

---

> > > > ### Author Response · Authors · 2022-11-18
> > > > **Thanks!**
> > > >
> > > > We are very glad that our responses can address your concerns. Thank you for your recognition of the idea of our work and your efforts on improving our work!

---

### Official Review · Reviewer_UNLu · 2022-10-25

**Confidence:** 3
**Correctness:** 3
**Technical Novelty And Significance:** 3
**Empirical Novelty And Significance:** 3
**Recommendation:** 6

**Clarity, Quality, Novelty And Reproducibility:**

The text is generally readable, but there are some issues with clarity that are distracting.

For example, the abstract states that one of the problems with planning in continuous control is that there is a “temporal dependency between actions in different timesteps.” It is not clear to me what this means or why it is true. This could be clarified. And later the text states that “the temporal dependency between actions implies that the action update in previous timesteps can influence the later actions.” I think this might be saying the same thing, but it needs some clarification as well.

Some descriptions of how the algorithm components fit together are hard to parse. For example, in the introduction, to help with potentially poor initializations of the planning process, the paper proposes to “to leverage the learned policy to provide the initialization of the action before planning and provide a conservative term at the planning to admit the conservation principle, in order to keep the small error of the learned model along the planning process.” I don’t understand what this is trying to say.

A description of the POMP algorithm is missing from the main text. How is this system actually used to help select actions at runtime? This is not clear to me from the text.

(Also see weaknesses above.)

The work presented appears novel.

The results in the paper could likely be reproduced using all of the information provided in the appendix, but I’m not sure if enough details exist in the main text for the results to be reproducible.


**Strength And Weaknesses:**

The main strength of this paper is the introduction of a novel algorithm based on DDP for planning in continuous control problems.

The main weakness of this paper is the presentation and discussion of the experimental results.

First, the experimental results compare the mean computed from 5 runs per algorithm. 5 samples is usually not enough to draw meaningful conclusions about the differences between algorithms. 10 runs would be better, and 30 would be ideal.

The shaded region in the plots is stated to be the standard deviation. Is this assuming the distribution of the learning curves is normal? Is this actually the case?

With only 5 runs it might actually be clearer just to plot all the runs individually instead of the mean. This would give the reader a better idea of the distribution and relative performance.

In the section “Is planning necessary to make a better decision in continuous control?” a new parameter N_p is introduced without any reference in the main text. Is this a parameter of MAAC? What does it mean? This seems to be central to understanding this section.

The section “How the learned model quality affect decision-making?” talks about how the amount of training data used to train the model affects planning utility. What is this model? How is it trained?

The text is missing some information to make Figure 4 more understandable. First plots 4.a and 4.b show “Improvements” on the y-axis. What is this? How is it computed. In 4.a what is policy quality? How is it measured? In 4.b what is model quality? How is it measured? Are they averages of something? A single sample?

I think there is some potentially interesting content in the ablation experiments, but in its current form it is hard to understand. It deserves a more thorough explanation and improved clarity.


**Summary Of The Paper:**

This paper proposes a new algorithm called Policy Optimization with Model Planning (POMP), for continuous control problems that incorporates a model-based planner derived from differential dynamic programming (DDP). DDP cannot be practically applied directly as a planner since it has a high computational cost and requires a known model. Thus, as a practical approximation, the paper presents Deep Differentiable Dynamic Programming (D3P), and proves its convergence. Experimental results show that POMP, which contains D3P as the planner, outperforms some model-free and model-based baselines on some simulated robotics continuous control tasks.

**Summary Of The Review:**

The paper presents a new model-based planner for continuous control problems that might be of interest to RL practitioners. However, some issues with clarity, and the presentation and discussion of the experimental results take away from the overall impact of this paper. Therefore, I argue to reject.

The paper could be improved by increasing the clarity throughout, and carefully expanding the discussion of the experimental results.

---

> ### Author Response · Authors · 2022-11-14
> **Response to Reviewer UNLu [Part 1]**
>
> Thanks for your recognition of our paper’s insight. Also, thank you very much for the valuable comments. It seems that most of your concerns are misled by the presentation. We have fixed them in the revised version of the manuscript. Here are our responses to your concerns.
>
> > **For the concerns about the number of experimental runs.**
>
> **A**:  Please note that our implementation is mainly based on two well-known baseline works MBPO [1] and MAAC [2], and we use the same number of trials, 5 runs as in [1][2][3]. For your concern, we have added 5 more seeds for our method in Figure 1 and use a t-test to test the hypothesis of whether the mean performance of our method is significantly better than the baseline method (MAAC) [2].  The results of the t-test show that the performance of our method is indeed significantly better than MAAC (p-value < 0.05 is a commonly used threshold to show the significance):
>
> |Env Name| Hopper | Walker | HalfCheetah | Ant | Humanoid |
> |---|---|---|---|---|---|
> |Env Step|75k|75k|75k|100k|100k|
> |pvalue|8.82e-5|0.00733|0.00036|0.00066|0.00157|
> |Env Step|100k|100k|100k|150k|150k|
> |pvalue|0.00861|0.01634|0.00159|6.56e-5|8.85e-7|
>
> > **For the concerns about the plot.**
>
> **A**: Thanks for your questions. We provide the following answers.
> - Please note that we plot the mean curve and the standard deviation in the shaded region because it is a commonly used visualization method in deep reinforcement learning works (e.g., A3C [4], PPO[5], TD3[6], SAC[7], MBPO[1], MAAC[2], and so on) and we do not take any assumption about the distribution.
> - Thanks for your suggestions about the individual visualization, we have added one figure that plots all 10 runs individually (Figure 9). This figure gives clearer information that our runs are robust with good performance.
>
> > **Meaning of $N_p$.**
>
> **A**: We have introduced $N_p$ in paragraph “Is planning necessary to make a better decision in continuous control?” of Section 5.2 and in Line 12 in Algorithm 2. It is the number of update times of the policy model after we collect 1 data from the real environment.  Yes, it is indeed an important parameter that describes the update frequency of the policy model w.r.t. the data collection. The experiment results show that only training policy model more times has no help or even negative influence on improving the decision performance.
>
> > **Questions about the “How does the learned model quality affect decision-making?”**
>
> **A**: “Model” means the neural network that is used to approximate the transition function and the reward function of the “environment”. We follow the routine usage of the terminology “model” in most model-based reinforcement learning. Since the model structure and the model learning is not the focus of this paper, we follow the MBPO [1] and MAAC [2] to use the same model structure and training procedure in our work. For a clearer understanding, we have added an introduction for the model structure and model training procedure in Appendix C.3 for self-contained and added a clear citation for the reference in our paper.
>
> > **Questions about the Figure 4.**
>
> **A**: Thanks for your questions. We provide the following answers.
> - “Improvements” means the evaluation return that uses our D3P planner subtracts the return that runs with only the policy model and without our D3P planner. Thus, the higher the improvements, the more effective our D3P planner is.
> - The “model quality” means the number of training data used to train the model. The “policy quality” means the average episode return of the policy when applying the policy in the environment.  We have revised the figure illustrations and the paper to make it clearer and easier to understand.
> - Specifically, “average return score” is illustrated as the x-axis for figure 4(a), and “training data number” is illustrated as the x-axis for figure 4(b). They are measured as follows. First, for each model and each policy, we evaluate the return using 10 trajectories. Then, we cluster the learned model and policy according to their training data number and the average return and then calculate the average performance improvements in each cluster.
>
> > **Unclear ablation experiments.**
>
> **A**: We have revised the paper about the ablation studies and more other study experiments. Kindly check the new version please. We hope it is much clearer now, if you have more unclear questions, we are welcome for further discussions.

---

> > ### Author Response · Authors · 2022-11-14
> > **Response to Reviewer UNLu [Part 2]**
> >
> > > **Questions about “temporal dependency between actions in different timesteps”.**
> >
> > **A**:  Thanks for your questions. We provide the following answers.
> > - Overall speaking, we have provided the intuition explanation and the theoretical results in our paper that can help to convince you about what that means and why it is correct.
> > - We first used a paragraph in our paper (see the paragraph above Equation (2) and the last paragraph in Section 4.1) to explain the intuition: Since we are optimizing the action sequence along a trajectory, the action update will change the trajectory sequentially. For example, the change of the action $a_t$ at $t$-step will influence the next state $x_{t+1}$ at $t+1$ step, and the corresponding action $a_{t+1}$ is then based on state $x_{t+1}$. Hence, clearly, $a_t$ will impact $a_{t+1}$.  Given our objective is a function of state and action, the different states will lead to the different optimal actions. Therefore, if we do not consider the state change due to the action update in the previous timesteps, the action update direction will not be toward the true gradient direction. Besides, the influence is proportional to the magnitude of the state change which is determined by the system property ($\frac{df}{dx}, \frac{df}{da}$) and previous action update $\delta a_{t-1}$.
> > - Besides, we also presented a corollary 1 in our paper (see Section 4.1). This corollary shows the necessity of considering the temporal dependency in the optimization process. The corollary shows that if we do not consider the temporal dependency between actions in different timestep, or in other words $\delta x=0$, the convergence rate will be slower than Equation (12) by introducing an extra error term.
> >
> > > **Questions about how the algorithm components fit together.**
> >
> > **A**:  Thanks for your questions. We provide the following answers.
> > - Please kindly check Section 4.2 and Appendix B and C for the detailed description of each part of the algorithm. We have revised them to make them clear to understand.  “Use the policy network and learned model to initialize the state-action trajectory” is describing Line 3 in Algorithm 2 in Appendix B. It means that the initial action used by the D3P planner is the output of the learned policy net. Specifically, before running the D3P planner, we first use the policy net to output the action at each time step. Then we can use the D3P planner to optimize these actions to make a better decision.
> > - “Provide a conservative term at the planning to admit the conservation principle” corresponds to the formulation in the last paragraph of Section 4.2.  We add an additional regularization term $\alpha \log P(x_i, a_i)$ to regularize the search space of the D3P planner. The term is used as an auxiliary reward, and we add it to the output of the reward function when doing planning in the evaluation phase. The term “conservative” is commonly used in offline RL which usually constrains the learned policy to be closer to the behavioral policy (such as Conservative Q-Learning [8]).
> >
> > > **Questions about the POMP algorithm.**
> >
> > **A**: Due to space limitations, we have to put the algorithm into the appendix and leave a brief description in the main text.
> >
> > > **Questions about the details in our main paper.**
> >
> > **A**: We put the most important descriptions and the experiments in the main text, while leaving others in the appendix, and this is to give a clear understanding of the overall picture of our work for most of the readers. For those who want to know more about the details, they can find the necessary information in the appendix.

---

> > > ### Author Response · Authors · 2022-11-14
> > > **Response to Reviewer UNLu [Part 3]**
> > >
> > > [1].	Janner, Michael, Justin Fu, Marvin Zhang, and Sergey Levine. "When to trust your model: Model-based policy optimization." Advances in Neural Information Processing Systems 32 (2019).
> > >
> > > [2].	Clavera, Ignasi, Yao Fu, and Pieter Abbeel. "Model-Augmented Actor-Critic: Backpropagating through Paths." In International Conference on Learning Representations. 2019.
> > >
> > > [3].	Wang, Tingwu, Xuchan Bao, Ignasi Clavera, Jerrick Hoang, Yeming Wen, Eric Langlois, Shunshi Zhang, Guodong Zhang, Pieter Abbeel, and Jimmy Ba. "Benchmarking model-based reinforcement learning." arXiv preprint arXiv:1907.02057 (2019).
> > >
> > > [4].	Mnih, Volodymyr, Adria Puigdomenech Badia, Mehdi Mirza, Alex Graves, Timothy Lillicrap, Tim Harley, David Silver, and Koray Kavukcuoglu. "Asynchronous methods for deep reinforcement learning." In International conference on machine learning, pp. 1928-1937. PMLR, 2016.
> > >
> > > [5].	Schulman, John, Filip Wolski, Prafulla Dhariwal, Alec Radford, and Oleg Klimov. "Proximal policy optimization algorithms." arXiv preprint arXiv:1707.06347 (2017).
> > >
> > > [6].	Dankwa, Stephen, and Wenfeng Zheng. "Twin-delayed ddpg: A deep reinforcement learning technique to model a continuous movement of an intelligent robot agent." In Proceedings of the 3rd International Conference on Vision, Image and Signal Processing, pp. 1-5. 2019.
> > >
> > > [7].	Haarnoja, Tuomas, Aurick Zhou, Kristian Hartikainen, George Tucker, Sehoon Ha, Jie Tan, Vikash Kumar et al. "Soft actor-critic algorithms and applications." arXiv preprint arXiv:1812.05905 (2018).
> > >
> > > [8].	Kumar, Aviral, Aurick Zhou, George Tucker, and Sergey Levine. "Conservative q-learning for offline reinforcement learning." Advances in Neural Information Processing Systems 33 (2020): 1179-1191.

---

> > ### Comment · Reviewer_UNLu · 2022-11-16
> > **Reply**
> >
> > Thank you for answering my questions and providing clarifications.
> >
> > RE: Result significance and plots
> >
> > You say that "we do not take any assumption about the distribution." But, conducting a t-test explicitly assumes that your samples are coming from a normal distribution. And in your plots, you plot the sample standard deviation as the shaded region. This is ostensibly the sample standard deviation of a normal distribution. So when you are comparing the performance of two algorithms in your plots, you are comparing the summary statistics of two distributions that are assumed to be normal. Are you sure that the distributions are normal? Not skewed or multi-modal? If your number of samples is large enough (usually n >= 30), the distribution of the sample means will be approximately normally distributed (central limit theorem).
> >
> > RE: Meaning of N_p
> >
> > The addition to the text clarifies the meaning of N_p. However, I do still find it a bit strange that you reference a parameter that is not introduced as part of an algorithm described in the main text (it is part of Algorithm 2 in the appendix).

---

> > > ### Author Response · Authors · 2022-11-17
> > > **Response to Reviewer UNLu [Part 3]**
> > >
> > > Thanks for your reply and suggestions to improve our work. Here are our responses to your concerns.
> > >
> > > > **Result significance and plots**
> > >
> > > **A**: For this question, we provide the following answers.
> > >
> > > 1. We have run 10 seeds for each environment task in Figure 1, and per your request, now we are running the remaining 20 seeds for each task. As we know, RL tasks are always computationally costly. Thus, we need a huge computational resource to complete the experiments (20*6=120 experiments). Considering the deadline for the Discussion stage 1 is approaching, we promise that we will update these results after the rebuttal.
> > > 2. Again, we would like to clarify that plotting the mean curve and the deviation in the shaded region is a commonly used visualization in many well-known RL literature, such as 5 runs in A3C[4],  3 runs in PPO[5], 10 runs in TD3, 5 runs in SAC[7], 5 runs in MBPO[1], 4 runs in MAAC[2]. We do not hold any assumptions, and we just follow the common practice.
> > > 3. Due to the space limitation, we have drawn the curve of 10 individual runs in Figure 9. We will update it to the 30 individual runs after we finish the extra 20 seeds experiments.
> > >
> > > >**Meaning of N_p**
> > >
> > > **A**: Per your request, we have moved Algorithm 2 to the main text. Please have a check. Now, the notation N_P is first introduced in Algorithm 2 and then explained in the first paragraph on page 8 (the first time it appears in the main paper).
> > >
> > > Thanks again for your reply.

---

> > > > ### Comment · Reviewer_UNLu · 2022-11-17
> > > > **Reply**
> > > >
> > > > Striving to have 30 runs will certainly improve the results in the paper. It is ok to have fewer runs when computation is a concern, but it is important to acknowledge the limitations and assumptions of the comparisons; arguably the field as a whole does not do a good job of this.
> > > >
> > > > I think having Algorithm 2 in the main text improves the presentation; it clarifies the parameters and the notation.

---

> > > > > ### Author Response · Authors · 2022-11-19
> > > > > **Response to Reviewer UNLu**
> > > > >
> > > > > Thanks for your reply and suggestions. Here are our responses to your last concern about the result plots.
> > > > >
> > > > > 1. We are trying our best to run 30 seeds for each task to improve the experimental results. Due to time and resource limitations, now we have finished 20 more seeds (a total of 30 seeds) for 3 environments and shown the results in Figure 11 in Appendix D.5. We can see that the impact of the number of experiment runs is limited to our method, which does not alter our experimental conclusion. **After finishing the remaining 3 environment tasks, we would update Figure 1 in the main paper with 30 seeds in our next version.**
> > > > > 2. We also add a paragraph in Appendix D.5: “Last, as the RL committee always shows the results with the mean and deviation values, we acknowledge that more runs of each task are needed to show robust and consistent experimental results for RL algorithms.”
> > > > >
> > > > > Thanks again for your reply.

---

> ### Comment · Reviewer_UNLu · 2022-11-21
> **After rebuttal**
>
> Based on the author's responses to my concerns, and their changes to the paper, I am going to raise my score to a borderline accept.

---

> > ### Author Response · Authors · 2022-11-22
> > **Thanks!**
> >
> > We are very glad that our responses can address your concerns. Thank you for your efforts on improving our work!

---

### Official Review · Reviewer_RrZx · 2022-10-28

**Confidence:** 3
**Correctness:** 3
**Technical Novelty And Significance:** 2
**Empirical Novelty And Significance:** 3
**Recommendation:** 8

**Clarity, Quality, Novelty And Reproducibility:**

Clarity: The paper is well written and the experiments and ablations are nicely done. As mentioned above, it would help clarity if the notation for the theoretical part is improved and additional details regarding the implementation, tasks etc are provided. Two additional relevant citations; [1] - an application of MCTS style planning to continuous control tasks, and [2] a related paper that uses sampling based MPC for high-dimensional continuous control tasks with learned models and a learned policy as a proposal distribution.

Quality & Novelty: The approach is built on top of prior work and specifically adds a Deep-DDP planner on top of existing work to showcase the strengths of model-based planning. Some of the presented contributions have been shown in prior work (iLQR/iLQG) as well and in some sense the approach seems to be a simplification of such methods to leverage NN based models. The presented results look quite promising and the approach can be quite useful if the strengths can be better quantified and the novelty made clear in the context of prior work.

Reproducibility: A link to an implementation on github is provided with the submission which helps with reproducibility.

[1] Springenberg, Jost Tobias, et al. "Local search for policy iteration in continuous control." arXiv preprint arXiv:2010.05545 (2020).

[2] Byravan, Arunkumar, et al. "Evaluating model-based planning and planner amortization for continuous control." arXiv preprint arXiv:2110.03363 (2021).

**Strength And Weaknesses:**

(+) The approach presents a promising algorithm for planning in continuous control spaces; the initial results in particular look promising on the tested tasks.

(+) The ablations provide good insight into the properties of the proposed method. I particularly like the sweep over model quality and proposal distributions in Fig. 4

(-) The clarity is a bit lacking, particularly it would help if the notation were simplified and more details are provided. Specifically details on the implementation are lacking: The approach seems to be using model ensembles, what are these used for? How are they used in the context of the planning algorithm? How are the gradients of the critic, model etc. needed for planning computed?

(-) More details on the tasks should be provided. Particularly the different embodiments are listed but it is not mentioned what the actual tasks are, is it walking/running/standing in place?

(-) The idea of feedback to handle the temporal nature of planning is not new. Approaches such as iLQR [1] and iLQG [2] had a feedback term. These papers should be cited and would be good to mention what the differences are compared to these -- in fact the major difference to me seems to be a simplification of the gradient computation from these methods & use of NN models. Please clarify this explicitly.

(-) It is surprising (and a bit disappointing) that such a small planning horizon (H <= 4) is sufficient for the proposed approach. Does the approach work with longer horizons or does it fail due to model approximation errors? What about a simple 1-step lookahead (H=1)? An ablation of the horizon is crucial to understanding the strength of the approach.

(-) Another ablation that is crucial is the choice of the planner. Fig. 3 shows an ablation replacing D3P with SGD but in general simple derivative based planners are poor choices for planning with neural networks. Is it instead possible to replace it with a simple sampling based planner, e.g. random shooting? For the short horizons (H <= 4) considered in the paper even such planners might work. It would be helpful to quantify this as currently it is not obvious if the improvement is due to the ability to do lookahead search (which can be achieved with many planners) vs the specific use of D3P.

[1] Li, Weiwei, and Emanuel Todorov. "Iterative linear quadratic regulator design for nonlinear biological movement systems." ICINCO (1). 2004.

[2] Todorov, Emanuel, and Weiwei Li. "A generalized iterative LQG method for locally-optimal feedback control of constrained nonlinear stochastic systems." Proceedings of the 2005, American Control Conference, 2005.. IEEE, 2005.

**Summary Of The Paper:**

This paper presents a novel model-based RL algorithm for continuous control tasks that leverages a planner inspired by the Differential Dynamic Programming (DDP) approach to generate actions during environment interactions. The planner, Deep DDP or D3P, computes a locally quadratic approximation of the planning objective leveraging the Bellman equation and computes a "delta" action update on top of the current update by optimizing this approximation. This is combined with a term that accounts for the feedback from prior action updates to correct the current action update. Additionally, actions are initialized using a learned policy network, and the transition model, critic and reward function used for D3P are also jointly learned. The transition model and rewards are trained in a supervised manner on collected data (both from random interactions & learned policy/planner output); the policy and critic are trained on a combination of real interactions and imagined data from the model. The approach is tested on six continuous control tasks using the Mujoco simulator and shows good results compared to baselines; several ablations are also provided.

**Summary Of The Review:**

Overall, this paper presents a simplification of the DDP algorithm to leverage neural networks for planning in continous control; this is combined with an RL approach for data generation and shows good performance across the tested tasks. Several strengths and weaknesses were called out; in particular additional ablations are needed to better quantify the strengths of the approach and the clarity of the text can be improved.

---

> ### Author Response · Authors · 2022-11-14
> **Response to Reviewer RrZx [Part 1]**
>
> Thank you for your positive comments on our work and thank you for your recognition of the paper’s clarity. We have revised our paper according to your suggestions and provide our response to address your concern below.
>
> > **Questions about the model structure and the gradient computing.**
>
> **A**: Thanks for your questions. We provide the following answers.
> - Since the model structure and model learning is not the focus of this paper, we follow the MBPO [1] and MAAC [2] to design our model structure and model learning procedure. In other words, the ensemble model structure is the same as MBPO and MAAC.  The motivation for using such an ensemble model is “a proper handling of both of uncertainties allows for asymptotically competitive model-based learning.”, which is stated in these works.
> - These models are used in Lines 7 and 8 in Algorithm 1 (D3P algorithm).
> - When doing the first-order planning, we need to calculate the gradient of the return w.r.t the action and state. Briefly speaking, we first forward rollout using our learned model and then backpropagate through the model to calculate the gradient. For example, to calculate the $f_x$ and $f_a$, we start from a state $x$ and an action $a$ and then select a model uniformly to do forward through the model. For backpropagation, we use the auto-differentiation of neural networks (i.e., [functorch](https://pytorch.org/functorch/stable/) ) to calculate the gradient or Jacobian matrix of the output of the model ($s’$) w.r.t. its input ($s,a$) along the computational graph that is constructed in the forward rollout phase.
>
> > **Concerns about the details of tasks.**
>
> **A**: Thank you for your suggestion. To make it clearer, we have added a section in Appendix C.2 to introduce these tasks. Since MuJoCo [3] is a well-known benchmark environment in the RL community, we put the descriptions for these tasks in Appendix C.2. More details can be also found in [3].
>
> > **Concerns about the related work.**
>
> **A**: Thank you for your suggestions. We have already cited the papers you mentioned in the main paper (Section2) and clarified the connection and the difference between our method and these methods in detailed related work section in Appendix A (we put them in the appendix due to space limitations).
>
> > **The difference between our D3P planner and iLQR.**
>
> **A**: Thanks for your questions. We provide the following answers.
> - We have revised our related work section in the main paper and make a detailed discussion about them in Appendix A.
> - Since both iLQR and our D3P planner are motivated by DDP method, they look similar naturally. But we should clarify that our method has several differences compared with iLQR, and these differences are well-designed to incorporate the neural network model.   (1) Computing the second-order derivative of the neural network based model will be computationally costly (Hessian matrix). In our method, we only rely on the first-order derivative of the model. (2) The iLQR method uses the second-order Talyor expansion of the Q-value function to handle the local optimization problem. But it is hard to guarantee that the hessian matrix is a negative definite matrix, which is a necessary condition for convergence. Here, we construct an auxiliary target function $D$ and use a first-order Taylor expansion for the Q function inside of the $D$ function to guarantee the non-positive definite matrix.
>
> > **Concerns about the planning horizon.**
>
> **A**: We fix the planning horizon $H$ to be the same as those in MAAC [2] since they have systematically studied this hyper-parameter in Section 5.2 in MAAC:   “The gradient error scales poorly with the horizon, and large horizons are detrimental since it magnifies the error on the models”. For your concern, we have added an ablation study in Appendix D.3 to show how the planning horizon $H$ influences the performance of our method, the result is shown in Figure 8. The results are consistent with prior work [1][2].

---

> > ### Author Response · Authors · 2022-11-14
> > **Response to Reviewer RrZx [Part 2]**
> >
> > > **Concerns about the sampling-based planning method.**
> >
> > **A**: Thanks for your questions. We provide the following answers.
> > - According to your question, we have added two experiments using the random shooting method and the cross-entropy method as the planner in the updated figure 3. We can see that our method is significantly better than the random shooting and the cross-entropy method, and the result shows the advantage of our first-order method compared to zero-order method.
> > - We have added an experiment to use MCTS + continuous UCT method in Appendix D.2 for a comparison. The results in Figure 7 show these methods are not effective in the continuous control and high dimensional setting, which implies that employing Muzero in continuous domain effectively is non-trivial. Since UCT is a principled way to do the exploration in the discrete domain, combining its idea with the D3P planner for continuous domain will be an interesting research direction in the future.
> > - Thanks for providing more relevant works, we have carefully read these papers and cited them properly. In a short summary, we have revised our related work section in both the main paper and Appendix A according to your comments and added more studies. Hope they can help address your concerns.
> >
> > [1]. Janner, Michael, Justin Fu, Marvin Zhang, and Sergey Levine. "When to trust your model: Model-based policy optimization." Advances in Neural Information Processing Systems 32 (2019).
> >
> > [2]. Clavera, Ignasi, Yao Fu, and Pieter Abbeel. "Model-Augmented Actor-Critic: Backpropagating through Paths." In International Conference on Learning Representations. 2019.
> >
> > [3]. MuJoCo， Advanced physics simulation. https://mujoco.org/

---

> > > ### Comment · Reviewer_RrZx · 2022-11-19
> > > **Response**
> > >
> > > Thanks to the authors for their detailed response, most of my comments have now been addressed with the additional ablation experiments and changes to the text. The paper also looks quite good in my opinion so I will change the score to an accept.

---

> > > > ### Author Response · Authors · 2022-11-21
> > > > **Thanks!**
> > > >
> > > > We are very glad that our responses can address your concerns. Thank you for your recognition of the idea of our work and your efforts on improving our work!

---

### Author Response · Authors · 2022-11-14
**General Comments and Revision Summary**

We thank all reviewers for your careful review and constructive comments and suggestions. We also thank all reviewers for your recognition of the idea, insights, and presentation of our work. We summarize our main revision of the paper below.

- According to the reviewer’s suggestion, we added several experiments, including the Random shooting method, the Cross-entropy method, and the MuZero + Continuous UCT (Progressive Widening) method.
- We carefully read the related works provided by all reviewers and cite them properly in the paper. We made a discussion to show the connection and difference between our method and the related work.
- We revised the Experiment section to make it easier to follow. Specifically, (1) we provided more experiment details in the main paper and the appendix. (2) We clarified several notations used in the ablation study.
- We revised the whole paper to make it easier to follow and fixed the typos.
- Last but not least, we thank all reviewers again for the effort they put into this paper which makes this paper better.

---

### Decision · Program_Chairs · 2023-01-20

**Decision:**

Accept: poster

**Justification For Why Not Higher Score:**

The reviewers felt the approach, while solid and well-evaluated, was not particularly innovative. One of the reviewers also felt there remained some clarity issues even after the rebuttal.

**Justification For Why Not Lower Score:**

The reviewers all agreed that this was a good paper with an interesting method and impressive results.

**Metareview: Summary, Strengths And Weaknesses:**

This paper introduces a gradient-based planning algorithm, D3P, designed specifically to be used in the context of deep learning for continuous control. D3P is leveraged in a deep learning agent called POMP and shows improvements over several baselines on Mujoco continuous control tasks.

The reviewers cited several strengths of the paper, including the algorithm being promising and the experimental evaluation being thorough. In the original submission, the reviewers flagged some issues with clarity, lack of comparisons to sample-based baselines like Random Shooting/CEM or search baselines like MCTS, some missing discussion of prior literature, and a number of other smaller issues. The authors provided a very thorough rebuttal including all requested baselines as well as additional discussion and clarifications in the paper. The reviewers were overall satisfied with these revisions and ultimately found this to be a strong paper worth presenting at ICLR. I therefore recommend acceptance.

**Note From Pc:**

if the above contains the word "oral" or "spotlight" please see: "oral" presentation means -> notable-top-5% and "spotlight" means -> notable-top-25%. As stated in our emails, we are disassociating presentation type from AC recommendations

**Summary Of Ac-Reviewer Meeting:**

N/A